# OmniJARVIS
# Unified Vision-Language-Action Tokenization Enables Open-World Instruction Following Agents

**Zihao Wang**[1], **Shaofei Cai**[1], **Zhancun Mu**[2], **Haowei Lin**[1], **Ceyao Zhang**[3], **Xuejie Liu**[1]
**Qing Li**[3], **Anji Liu**[4], **Xiaojian Ma**[3], **Yitao Liang**[1*]
**Team CraftJarvis**

[1]Institute for Artificial Intelligence, Peking University
[2]Yuanpei College, Peking University
[3]Beijing Institute for General Artificial Intelligence (BIGAI)
[4]University of California, Los Angeles
`{zhwang,caishaofei}@stu.pku.edu.cn`
`xiaojian.ma@ucla.edu,liuanji@cs.ucla.edu,yitaol@pku.edu.cn`

## Abstract

This paper presents `OmniJARVIS`, a novel Vision-Language-Action (VLA) model for open-world instruction-following agents in Minecraft. Compared to prior works that either emit textual goals to separate controllers or produce the control command directly, `OmniJARVIS` seeks a different path to ensure both strong reasoning and efficient decision-making capabilities via *unified* tokenization of **multimodal interaction data**. First, we introduce a *self-supervised* approach to learn a behavior encoder that produces discretized tokens for behavior trajectories $\tau = \{o_0, a_0, \dots\}$ and an imitation learning policy decoder conditioned on these tokens. These additional *behavior tokens* will be augmented to the vocabulary of pretrained Multimodal Language Models. With this encoder, we then pack long-term multimodal interactions involving task instructions, memories, thoughts, observations, textual responses, behavior trajectories, *etc*. into unified token sequences and model them with autoregressive transformers. Thanks to the semantically meaningful behavior tokens, the resulting VLA model, `OmniJARVIS`, can reason (by producing chain-of-thoughts), plan, answer questions, and act (by producing behavior tokens for the IL policy decoder). `OmniJARVIS` demonstrates excellent performances on a comprehensive collection of atomic, programmatic, and open-ended tasks in open-world Minecraft. Our analysis further unveils the crucial design principles in interaction data formation, unified tokenization, and its scaling potentials. The dataset, models, and code will be released at `https://craftjarvis.org/OmniJARVIS/`.

## 1 Introduction

Upon the success of pretrained Large Language Models (LLMs) [7, 35, 40, 17, 13] and Multimodal Langauge Models (MLMs) [31, 20, 1, 53, 33], some recent works have been venturing into developing Vision-Language-Action (VLA) models [6, 22, 47, 38], a promising pathway towards the ultimate goal of building autonomous agents that can follow and even self-generated instructions to fulfill various reasoning and acting tasks in open world environments. Among them, two most prominent

---

[*]Corresponding Author.

38th Conference on Neural Information Processing Systems (NeurIPS 2024).

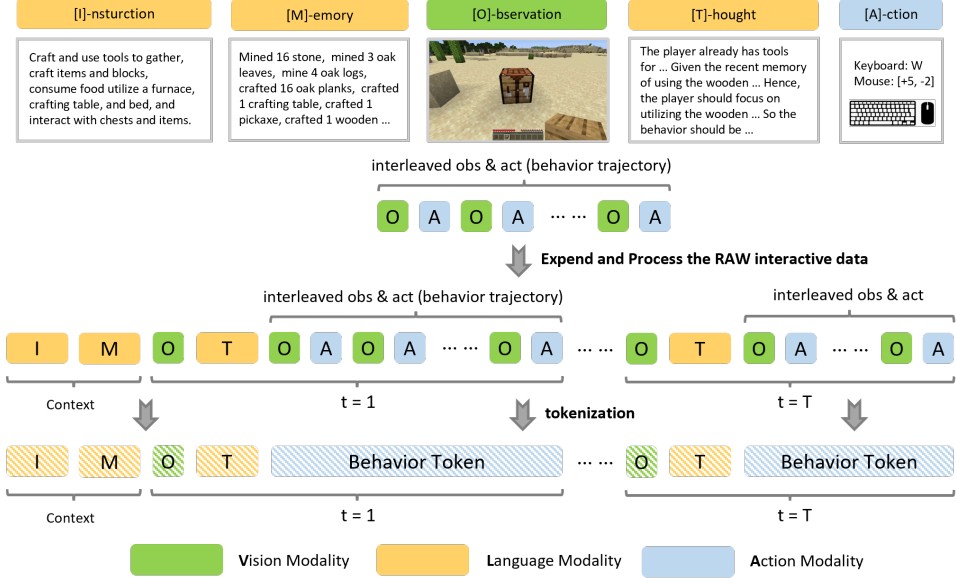

Figure 1: **Illustration of multi-modal interaction data for decision-making.** A canonical interaction sequence depicting the human decision-making process starts from a given task instruction and memory, followed by a series of sub-task completion which involves initial observations, chain-of-thought reasoning, and behavior trajectories. Our proposed VLA model OmniJARVIS jointly models the vision (observations), language (instructions, memories, thoughts), and actions (behavior trajectories) as **unified** autoregressive sequence prediction. A *self-supervised* behavior encoder (detailed in Section 2 and Figure 2) converts the actions into behavior tokens while the other modalities are tokenized following the practices of MLMs [31, 3, 1].

architectures have been proposed: 1) Combining an off-the-shelf MLM [31, 1] with separate goal-conditioned controllers [28, 10, 9], where MLM reasons, plans and pilots the controllers by producing textual goal instructions, *e.g.* DEPS [46], JARVIS-1 [47], voyager [44]; 2) Tuning a pretrained MLM into producing control commands directly, while maintaining the reasoning and language capabilities, *e.g.* RT-2 [6], LEO [22]. However, these two designs could still have significant drawbacks when it comes to open-world environments. First, an open world (e.g., Minecraft) usually teams up with an infinite number of complex and highly contextualized tasks [16, 29], and it can be fairly challenging to depict them in text only. Therefore, VLA models that solely depend on text to communicate with the text-conditioned policies [47, 46] may fail to correctly pilot these controllers. On the other side, emitting the control command directly [6, 22] without invoking separate controllers could alleviate the aforementioned communication problem but given the long-horizon nature of open-world tasks, it is less practical to perform long-term control with a large VLA model as the context length requirement, computation cost and inference efficiency could become unaffordable.

In this paper, we aim to tackle the aforementioned issues of existing VLA models when facing open-world environments: **complex & context-dependent tasks** and **long-term tasks**. Our **key insight** originates from the observation of human decision-making: Given these open-world tasks, humans can make informed decisions via multi-round mental, verbal, and physical interactions (an illustration can be found in Figure 1). Therefore, if the VLA model can manage to learn from such interaction data, it may master the underlying human decision-making procedures. However, modeling interaction data is *non-trivial*: it is **multi-modal**, encloses vision (mostly observations), language (instructions, thoughts, etc.), and actions (behavior trajectories). Compared to the fruitful explorations on jointly tokenizing vision and language [31, 3, 43, 1] into sequences for autoregressive modeling [7], tokenizing behavior trajectories (actions) is hard due to the following reasons. On the one hand, directly using low-level actions from the environment would pose huge challenges to the model's ability to process long sequences, which significantly hurts performance. It also hinders us from leveraging the planning ability of generative models. On the other hand, language-level action tokens require significantly more supervision and cannot accurately describe all possible actions.

To this end, we propose OmniJARVIS, a novel VLA model that jointly models vision, language, and actions in interaction data with unified tokenization. OmniJARVIS comprises two **key ideas**: 1) **Behavior Tokenization.** We introduce a *self-supervised* approach to learn a behavior encoder

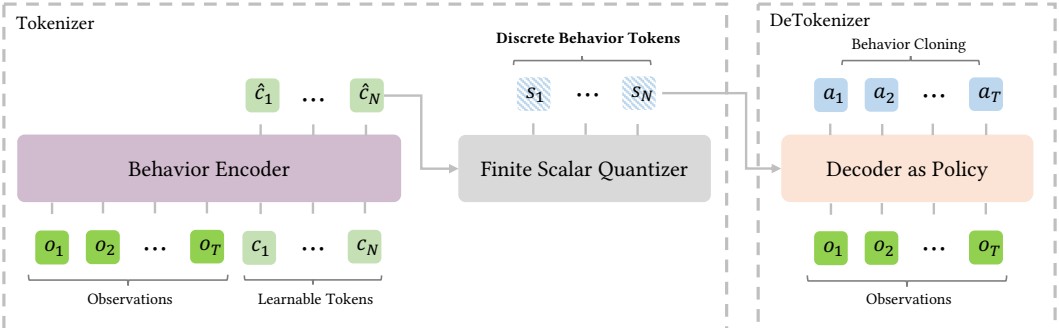

Figure 2: **Self-supervised learning for behavior tokenizer of** `OmniJARVIS`. We modify the VAE-based self-supervised learning of behavior trajectories in [10] to train the behavior tokenizer and de-tokenizer in `OmniJARVIS`. Specifically, we adopt the auto-encoding objective but replace the Gaussian latent with a discrete representation based on Finite Scalar Quantizer [34]. The encoder will then be used as the behavior tokenizer to produce discrete tokens from the actions (behavior trajectories) in multimodal interaction data, while the behavior tokens emitted by `OmniJARVIS` will be sent to the policy decoder to perform motor control.

that produces discretized tokens for actions (behavior trajectories) and an imitation learning policy decoder conditioned on these tokens (Section 2); 2) **Autoregressive Modeling.** By augmenting these *behavior tokens* into the vocabulary of pretrained MLMs, we pack the multimodal interaction data into unified token sequences and learn a transformer on these sequences with an autoregressive modeling objective. We conduct comprehensive evaluations in the open-world Minecraft Universe [29]. `OmniJARVIS` demonstrates impressive performances on a wide range of atomic, programmatic, and open-ended Minecraft tasks. Our analysis confirms several critical design choices in data formation, tokenization, and the scaling potential of `OmniJARVIS`. Our contributions are as follows:

- We propose `OmniJARVIS`, a novel VLA model capable of following instructions to reason, plan, and act in open-world environments by jointly modeling vision, language, and actions in multimodal interaction data for decision-making.

- We propose a self-supervised approach to learn a behavior encoder to tokenize actions and an imitation learning policy decoder to produce control commands from behavior tokens emitted by `OmniJARVIS`, allowing joint learning of VLA and smooth action readout.

- We conduct extensive evaluations in open-world Minecraft to demonstrate `OmniJARVIS`'s proficiency across various tasks and present in-depth analyses to reveal valuable insights.

## 2 A Tokenizer for Behaviors

As illustrated in Section 1, a key challenge for VLA is the mismatch between the action modality and other modalities such as the language instructions. A key insight is that a good amount of knowledge about the effects of actions can be learned directly from behavior trajectories $\{\tau^{(i)}\}_i$. We propose to learn a behavior tokenizer in addition to the well-studied vision and language tokenizers to achieve unified tokenization of the vision, language, and actions in multimodal interaction data (Figure 1). We pose two main requirements to the behavior tokens. First, they should be able to express complete and diverse behavior from (short) trajectories. Further, the tokens should contain semantic information so that they are compatible with the other modalities, which enables the reasoning and planning ability of LLMs (e.g., by conducting chain-of-thought reasoning).

Specifically, we aim at producing a set of $N$ discrete **behavior tokens** $s_1^{\text{bhv}}, \ldots, s_N^{\text{bhv}}$ from a behavior trajectory $\tau = \{o_0, a_0, \ldots\}$. Further, a de-tokenizer is needed to map these tokens back to an action rollout in the environment that reproduces the goal achieved in $\tau$. GROOT [10] explores a VAE-based approach to jointly learn a latent representation of behavior trajectories and an imitation learning policy decoder that conditions the latent as goal. However, the continuous latent cannot be used as the behavior tokens as they can be more difficult to learn and decode with the existing discrete tokens of pretrained MLMs [22, 32]. Therefore, we replace the Gaussian latent in GROOT with an improved vector quantized discrete latent called Finite Scalar Quantization (FSQ) [34]. We adopt a quantization configuration of $[8, 8, 8, 6, 5]$, which means a code with a length=5 and a codebook size of $8 \times 8 \times 8 \times 6 \times 5 = 15360$ is produced. The configuration is selected by a simple grid search. Overall, the behavior tokenizer (behavior encoder) $e_\phi(o_{1:T})$ and the de-tokenizer (IL policy decoder)

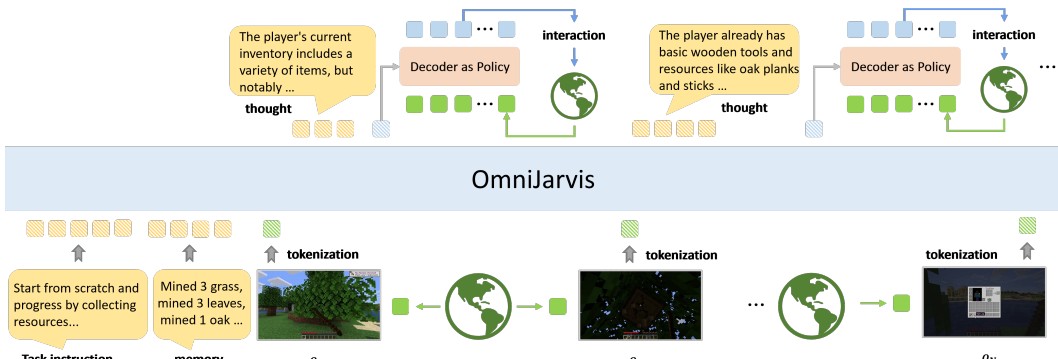

Figure 3: **Architecture and Inference of** `OmniJARVIS`**.** The main body of `OmniJARVIS` is a multimodal language model (MLM) augmented with additional behavior tokens. Given a task instruction, initial memory, and observation, `OmniJARVIS` will iteratively perform chain-of-thought reasoning and produce behavior tokens as a means of control via the decoder policy (behavior de-tokenizer). Every 128 steps, `OmniJARVIS` is forced to reason again and produce new behavior tokens with the latest observation. (Not shown above) `OmniJARVIS` can also make textual responses, *e.g.* answering questions.

$\pi_\theta(a_t|o_{1:t})$ is learned with the following objective:

$$\underset{(\phi,\theta)}{\operatorname{argmin}} \, \mathbb{E}_{\tau \sim \mathcal{D}} \left[ \sum_{t=1}^{T} -\log \pi_\theta(a_t|o_{1:t}, f(e_\phi(o_{1:T}))) \right], \tag{1}$$

where $f(\cdot)$ denotes the finite scalar quantizer. We choose a non-causal (bidirectional) transformer and a causal transformer to parameterize the encoder $e_\phi(o_{1:T})$ and the policy decoder $\pi_\theta(a_t|o_{1:t})$, respectively. In practice, we set $T = 128$ as the trunk size of the behavior trajectory to be encoded. We will discuss how to handle trajectories longer than 128 in the next section.

Compared to our behavior tokenization, most prior work in VLA models, either represents the behavior trajectories in interaction data as a textual goal description and invokes a separate goal-conditioned controller [47, 46], or represents the state-action sequence $\{o_0, a_0, \dots\}$ directly as in Decision Transformers (DT) [11, 22, 38, 6]. Our approach offers a more compact but still informative representation of the actions part in multimodal interaction data. Moreover, the action readout, *i.e.* simply sending the behavior tokens to the policy decoder, is also more efficient than the DT-style direct control from VLA models [38, 6, 22].

## 3 Multimodal Interaction Data and `OmniJARVIS`

As illustrated in Figure 1, canonical multimodal interaction data comprises vision (observations), language (instructions, memories, thoughts), and actions (behavior trajectories). However, it can be difficult to directly collect such interaction data from human annotators. Therefore, we propose to convert an existing Minecraft gameplay dataset [2] into the multimodal interaction data required by `OmniJARVIS`. We begin with a formal definition of the interaction data, followed by our approach for data conversion and augmentation from existing datasets, and finish up with the architecture, formulation of learning on such interaction data, and inference procedure of `OmniJARVIS`. An overview of `OmniJARVIS` architecture and inference can be found in Figure 3.

### 3.1 Data Formation

An interaction sequence of decision-making $\mathbb{D} = \{D_t\}_{t=0}^{T}$ comprises $T$ segments. Each segment $D_t$ can be a sentence of text words $\{w_i\}_{i=1}^{N}$, *i.e.* the language part such as instructions $D_t^{\text{inst}}$, memory $D_t^{\text{mem}}$ or thoughts $D_t^{\text{tht}}$. $D_t$ can also be an image $I$, *i.e.* the vision part such as observations $D_t^{\text{obs}} = I$. Finally, $D_t$ may belong to the action (behavior trajectory) part, *i.e.* $D_t^{\text{bhv}} = \{o_0, a_0, \dots\}$. We assume these segments follow the ordering below (Figure 1):

$$\underbrace{D_0^{\text{inst}}, D_1^{\text{mem}}}_{\text{Context}}, \underbrace{D_2^{\text{obs}}, D_3^{\text{tht}}, D_4^{\text{bhv}}}_{\text{sub-task 1}}, \underbrace{D_5^{\text{obs}}, D_6^{\text{tht}}, D_7^{\text{bhv}}}_{\text{sub-task 2}}, \dots \tag{2}$$

We tokenize such a sequence of segments into a series of tokens $\{s_0, \dots, s_M\}$ using the vision and language tokenizer offered by a pretrained MLM and the behavior tokenizer introduced in Section 2.

## 3.2 Preparing Multimodal Interaction Data

In reality, many segments of the multimodal interaction $\mathbb{D}$ can be missing in public datasets. We consider the Minecraft contractor data released by OpenAI [2] and it only contains behavior trajectories $D_t^{\text{bhv}}$. Therefore, we need to properly augment the data with the additional textual segments including instructions $D_t^{\text{inst}}$, memory $D_t^{\text{mem}}$, and thoughts $D_t^{\text{tht}}$. We follow the prior practices [22, 31] to synthesize the required text using LLMs. Below, we detail how each type of segment is constructed. More details can be found in *appendix*.

**Synthesis of instruction** $D_t^{\textbf{inst}}$**.** The instruction is a high-level description of what task is being performed in the current interaction sequence. The considered OpenAI Minecraft data includes *meta information* of each gameplay video, which depicts fundamental events that happened during in Minecraft gameplay, *e.g.* what block was just destroyed, what entity was just killed, what item was just crafted, *etc*. Such meta-information can provide a basic overview of what the player has been through in the gameplay. We therefore prompt an LLM into summarizing the gameplay with the meta information. The summary will be used as the instruction $D_t^{\text{inst}}$ of the current trajectory.

**Synthesis of memory** $D_t^{\textbf{mem}}$**.** The memory is the summary of what agents have finished in the previous interaction sequences. Due to the limited sequence length that the auto-regressive model can handle, the model needs to learn to summarize key information related to the task in historical interactions and ignore behaviors unrelated to instructions. The memory will be updated based on the results of each episode trunk and used for subsequent episode trunks. We therefore prompt an LLM into summarizing the gameplay with the meta information. The summary will then be used as the memory $D_t^{\text{mem}}$ of the current interaction trajectory. The memory prompt can be found in Appendix F.

**Synthesis of thought** $D_t^{\textbf{tht}}$**.** The thought is the agent's reasoning and explanation of its own decisions. Previous methods have confirmed that using thought-enhanced interaction data helps language models understand decision-making [21]. Compared to labeling thoughts by humans [50, 6], we assume that thought is an intermediate variable that can be determined by the actions taken and observations made before and after the action, which is similar to an Inverse Dynamics Model [2]. We therefore prompt an LLM into estimating the thought of decisions with in-context learning, which will then be used as the thought $D_t^{\text{tht}}$ of the current behavior. Details can be found in Appendix E.

## 3.3 Architecture, Training, and Inference of `OmniJARVIS`

As illustrated in Figure 3, `OmniJARVIS` is built upon a pretrained MLM. We augment the original vocabulary of the MLM with additional tokens from the behavior tokenizer. Specifically, as we adopted the $[a, b, c]$ FSQ configuration (Section 2), we augment with $a + b + c$ new tokens as each behavior comprises $n$ behavior tokens $s_1^{bhv}, \ldots, s_n^{bhv}$ corresponding to $n$ FSQ levels. We formulate the learning objective of `OmniJARVIS` following [7, 37] in a prefix language modeling fashion. For a batch $\mathcal{B}$ of token sequence $s$, we optimize `OmniJARVIS` via:

$$\mathcal{L}(\theta, \mathcal{B}) = -\sum_{b=1}^{|\mathcal{B}|} \sum_{t=1}^{T} \log p_\theta(s_{\text{res}}^{(b,t)} | s_{\text{res}}^{(b,<t)}, s_{\text{prefix}}^{(b,1)}, ..., s_{\text{prefix}}^{(b,L)}), \quad (3)$$

where $s_{\text{prefix}}$ denotes the prefix token, which is tokenized from the segments that served as context for reasoning and decision-making, *i.e.* instruction $D_t^{\text{inst}}$, memory $D_t^{\text{mem}}$ and observation $D_t^{\text{obs}}$ within the interaction sequence (Equation 2). The remaining tokens (tokenized from thought $D_t^{\text{tht}}$ and behavior trajectory $D_t^{\text{bhv}}$) will be predicted in an autoregressive fashion. From a high level, `OmniJARVIS` is trained to reason (producing thought tokens) and act (producing behavior tokens) from contexts with task instructions, memory, and current observations. During inference, we begin with the feeding `OmniJARVIS` with a task instruction, an empty memory, and an initial observation. `OmniJARVIS` will produce a chain-of-thought as a means of reasoning and subsequently, emit behavior tokens for control. Every $N$ steps, it is forced to reason again to produce new behavior tokens with the latest observation. We empirically set $N = 32$.

# 4 Capabilities and Analysis

## 4.1 Overview

**Training details and Datasets.** The training of the `OmniJARVIS` is divided into two stages. In the first step, we use a self-supervised training method to train a Behavior Tokenizer, including the Encoder and Decoder jointly. We use FSQ as a quantization method and build a codebook

with 8*8*8*6*5 discrete codes. The training data for Behavior Tokenizer comes from Contractor Dataset [2], which is a collection of Minecraft gameplay videos. The training parameters and details remain consistent with GROOT, which can be found in Appendix A.

In the second stage, we use this behavior tokenizer to process Minecraft offline trajectories to obtain behavior token sequences. We add 35 (8+8+8+6+5) additional tokens to the MLM tokenizer as behavior tokens for unified representation, so each time the VLA needs to output a continuous sequence of 5 tokens to represent a complete behavior. We use GPT-3.5 to synthesize thought, memory, and instruction to raw offline datasets to build complete interaction data. The specific prompt can be found in Appendix E. These data collectively constitute the embodied instruction-following dataset of `OmniJARVIS`, including 600k trajectories and about 900M tokens.

The training dataset of `OmniJARVIS` further includes a large amount of QA data about Minecraft. We generate a large number of seed questions about these texts using web pages on the Minecraft wiki. Then, we use the self-instruct method to generate a large number of creative questions and instructions. This constructed QA dataset consists of 300k conversations with about 90M tokens. During the training process, the QA data and instruction-following data are mixed, with a total of about 1T tokens, to train `OmniJARVIS`. In specific, we SFT (supervised finetune) LLaVA-7B [31]. The details can be found in Appendix A. To further demonstrate the generalizability of the method, we also fine-tune LLaVA at different scales and VLM Fuyu-8B with different architectures. The relevant results are presented in Section 4.5 and Section 4.6.

**Experimental Setups.** We conduct experiments in the complex and open-world environment of Minecraft, a voxel-based 3D video game that has garnered significant attention from real-life research due to its popularity and diverse mechanics [18, 16]. We first evaluate `OmniJARVIS` with atomic tasks, which are skill-level tasks, testing VLAs' ability to follow simple and straightforward instructions. Then we evaluate `OmniJARVIS` with programmatic tasks, which require the agent to obtain an item starting from an empty inventory. The success of these tasks requires VLAs to decompose the provided instruction into atomic-level subtasks, and hence tests VLAs' complex reasoning ability. Finally, we test `OmniJARVIS` with open-world embodied question-answering benchmarks and creative free-form instruction-following. We also conduct ablation experiments of `OmniJARVIS` with different behavior tokenizers, different training dataset formats, and different vision tokenizations. Finally, we explore the generalization abilities of `OmniJARVIS` of Atari Games and the scaling potential of `OmniJARVIS` with different models and data scales.

## 4.2 Main Results I: Short-horizon Atomic Tasks

Atom tasks are various simple skills that agents in Minecraft need to master. They are basic tasks yet are fundamental skills that agents need to master during the learning process. We first evaluate `OmniJARVIS` with our learned behavior tokenizer on these tasks.

We select "chopping trees" , "digging dirt" , "mining stones" , and "collecting wheat seeds"  as the evaluation tasks. We directly take those short task descriptions as instructions for agents. We use text-conditioned VPT [2], Open-world Control [9], STEVE-I [28], and video-instructed GROOT [10] as baselines. We compute the average rewards of different agents on every task in Table 1 across 10 runs. By observing the environment and adjusting action tokens dynamically, `OmniJARVIS` effectively follows straightforward instructions across various scenarios. It consistently achieves a high average reward with minimal standard deviation.

## 4.3 Main Results II: Long-horizon Programmatic Tasks

To further verify the ability of `OmniJARVIS` to complete tasks with long sequences, we use 30 programmatic tasks to evaluate the performance of different agents. These tasks require the agent to start from an empty inventory in a new world until obtaining the final required items, which is usually a chain of atom tasks. These tasks are divided into five groups based on difficulty: wooden, food, stone, iron, and diamond. For example, the prompt for task "Obtain a diamond pickaxe"  is "Give you nothing in the inventory, obtain a diamond pickaxe." This task requires more game time and more complex planning for up to 10 different intermediate items [2]. We list all programmatic tasks and its corresponding instructions in the Appendix C.1.

**Baselines** are divided into two types: 1) directly outputs actions, namely the native behavior tokenizer, including STEVE-I [28] and GROOT [10]. 2) using pretrained LLM as a planner to output language goals and connect the STEVE-I to execute these goals, including Zero-Shot Planner (GPT) [23],

Table 1: Evaluation results (rewards) on short-horizon atom tasks. The text-conditioned VPT [2] ("VPT (text)*") is from Appendix I of its paper.

| Method | Condition | ⬛↑ | 🟫↑ | ⬜↑ | 🔧↑ |
|---|---|---|---|---|---|
| VPT*[text] [2] | Language | $2.6^{\pm0.3}$ | $9.2^{\pm0.7}$ | - | $0.8^{\pm0.1}$ |
| STEVE-I [28] | Language | $11.0^{\pm3.0}$ | $10.0^{\pm2.5}$ | $3.2^{\pm1.6}$ | $5.1^{\pm2.5}$ |
| GROOT [10] | Video | $14.3^{\pm4.7}$ | $19.7^{\pm8.7}$ | $19.0^{\pm11.3}$ | $7.3^{\pm0.6}$ |
| OmniJARVIS | Language | $10.8^{\pm5.2}$ | $20.3^{\pm9.2}$ | $25.8^{\pm2.9}$ | $8.2^{\pm3.6}$ |

Table 2: Results on open-ended instruction following.

| Instruction Following ↓ | VPT | STEVE-1 | Voyager | DEPS | Ours |
|---|---|---|---|---|---|
| | 975.9 | 972.7 | 932.1 | 929.5 | 886.2 |

Table 3: Results on open-ended question answering. We use LLM-as-judge [35] to evaluate the accuracy.

| QA ↑ | Vicuna-7B | Vicuna-13B | LLaMA2-70B | GPT-3.5 | Ours |
|---|---|---|---|---|---|
| | 2.34 | 2.85 | 2.50 | 7.50 | 8.40 |

Table 4: Success rate of different agents on long-horizon programmatic tasks.

| Method | Action Tokenizer | Wooden (10) | Food (5) | Stone (5) | Iron (5) | Diamond (5) | Average |
|---|---|---|---|---|---|---|---|
| STEVE-I [28] | Native | $0.04^{\pm0.07}$ | $0.00^{\pm0.00}$ | $0.00^{\pm0.00}$ | $0.00^{\pm0.00}$ | $0.00^{\pm0.00}$ | 0.01 |
| GROOT [10] | Native | $0.05^{\pm0.08}$ | $0.00^{\pm0.00}$ | $0.00^{\pm0.00}$ | $0.00^{\pm0.00}$ | $0.00^{\pm0.00}$ | 0.02 |
| GPT [23] | Language | $0.26^{\pm0.14}$ | $0.08^{\pm0.04}$ | $0.24^{\pm0.05}$ | $0.04^{\pm0.05}$ | $0.00^{\pm0.00}$ | 0.15 |
| ReAct [50] | Language | $0.44^{\pm0.11}$ | $0.12^{\pm0.04}$ | $0.30^{\pm0.10}$ | $0.06^{\pm0.05}$ | $0.00^{\pm0.00}$ | 0.23 |
| DEPS [46] | Language | $0.78^{\pm0.11}$ | $0.12^{\pm0.04}$ | $0.68^{\pm0.08}$ | $0.16^{\pm0.05}$ | $0.04^{\pm0.05}$ | 0.43 |
| OmniJARVIS | FSQ GROOT | $0.95^{\pm0.07}$ | $0.44^{\pm0.05}$ | $0.82^{\pm0.08}$ | $0.32^{\pm0.11}$ | $0.08^{\pm0.04}$ | 0.59 |

ReAct [50], and DEPS [46]. We use success rate to evaluate the completion of tasks, that is, whether the task is completed within the specified time. The experimental results are listed in Table 6.

Programmatic Tasks usually require complex reasoning for planning. While STEVE-I and GROOT can only finish short skill-level tasks in atom tasks, is difficult to finish these programmatic tasks. Agents based on Language behavior tokenizer can complete complex tasks including diamond group ones, but with a low success rate. This is because these in-context learning methods leverage the pretrained LLM which may lack the necessary knowledge about this world. It is worth noting that in the Food group, agents based on Language Tokenizer have an average success rate of around 10%, as this set of tasks does not require complex reasoning. This indicates that Language-conditioned Tokenizers need additional language-conditioned trajectories as supervision for training while there was less such data available during STEVE-I's training phase leading to significant performance gaps. Meanwhile, OmniJARVIS uses a self-supervised trained behavior tokenizer which does not require extra language labels and hence receives more training resulting in good performance across a wider range of tasks. We will further prove this in the next set of Creative Task experiments.

### 4.4 Main Results III: Open-ended Question-Answering and Instruction Following Tasks

The open-ended tasks differ from programmatic tasks due to the lack of straightforward success criteria [16]. We select the long-term open-ended tasks which usually need at least 5 minutes of human-playing time to finish. The task prompts can be found in Appendix C.3. Following image generation and video generation tasks [19, 41], we take the Fréchet Sequence Distance (FSD) metrics to evaluate the correlation between agent rollout video and creative instruction. Specifically, we first ask human experts to finish the creative task prompts under randomly generated worlds and record the game-playing videos $V_{\text{human}}$. Then, we provided the task prompts for different Minecraft agents, and obtained a rollout video set $V_{\text{agent}}$. Similar to FID [19], we used MineCLIP [16] to calculate the embedding of video clips and computed FSD for the embedding distributions of human and agent rollout videos. The analysis of the metrics can be found in Appendix B.

We further conduct open-ended embodied question-answering benchmarks to evaluate the ability of the agent to complete open-ended instructions and grasp world knowledge. The questions answering instructions set can be found in Appendix C.2. The evaluation results can be found in Table 3 and Table 2. OmniJARVIS is the agent that can simultaneously complete both types of tasks and has achieved the best performance in different task sets, surpassing strong baselines including Voyager [44] and DEPS [46]. Also, it maintains strong reasoning capability, especially on embodied question answering compared to LLM baselines (with image captions as visual context).

### 4.5 Insights and Analysis

**Interactive Dataset Format.** We explore the crucial roles played by the different type of segments in interaction data, including the instruction, memory, thought, and caption tokens. The results can be found in Table 4, where we evaluate the loss on predicting the behavior tokens. It can be seen that instruction and thought can be more critical to the successful prediction of behavior tokens. This is consistent with our hypothesis – making informed decisions requires task instruction and reasoning.

Figure 4: Ablation experiments on `OmniJARVIS` with different behavior tokenizers, vision tokenizers, and training on different interactive datasets. The first line is training on the unconditional interactive dataset, i.e., without instructions on the trajectories. `OmniJARVIS` with VQ-GROOT [42, 10] shows no results because of training collapse.

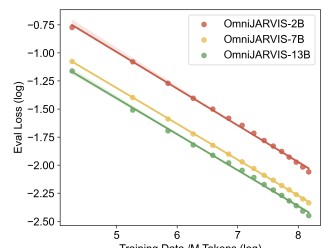

Figure 5: **Scaling potential of** `OmniJARVIS`. Its evaluation loss continues to drop with the growth of data and model parameters. The Pearson coefficients for the 2B, 7B, and 13B models are 0.9991, 0.9999, and 0.9989.

| Behavior Tokenizer | Vision Tokenizer | Dataset Format | | | | Loss | |
|---|---|---|---|---|---|---|---|
| | | Instruction | Caption | Thought | Memory | Train | Eval |
| | | ✗ (unconditional) | | | | 0.33 | 0.67 |
| | | ✓ | ✗ | ✗ | ✗ | 0.46 | 0.51 |
| FSQ GROOT | LLaVA | ✓ | ✓ | ✗ | ✗ | 0.44 | 0.48 |
| | | ✓ | ✓ | ✓ | ✗ | 0.32 | 0.33 |
| | | ✓ | ✓ | ✓ | ✓ | 0.16 | 0.17 |
| FSQ GROOT | Captioner+ | ✓ | ✓ | ✗ | ✗ | 0.49 | 0.52 |
| FSQ GROOT | FUYU | ✓ | ✓ | ✗ | ✗ | 0.42 | 0.44 |
| GROOT | LLaVA | ✓ | ✓ | ✓ | ✓ | 0.44 | 0.48 |
| VQ GROOT | LLaVA | ✓ | ✓ | ✓ | ✓ | - | - |

Table 5: Ablation experiments on behavior tokenizer with different code vocabulary size.

| Codebook size | FSQ Levels | Training Iterations | Train Loss | Eval Loss | Reconstruction FSD ↓ | Sampling FSD ↓ | Average Rewards ↑ | Codebook Usage |
|---|---|---|---|---|---|---|---|---|
| e8 | [8,6,5] | 180k | 2.746 | 3.161 | 46.57 | 68.90 | $0.63^{\pm0.67}$ | 93.75% |
| e10 | [8,5,5,5] | 180k | 3.011 | 3.148 | 43.67 | 61.85 | $0.54^{\pm1.21}$ | 97.65% |
| e14 | [8,8,8,6,5] | 240k | 3.092 | 3.116 | **42.72** | **57.37** | $2.27^{\pm2.45}$ | 92.36% |

**Vision Tokenization.** We also evaluate training `OmniJARVIS` with different vision tokenization, including ImageCaptioner + LLaMA2-7B [12, 40] (basically converting the vision input into textual captions), fuyu-8b [3], and LLaVA-7B [31] architecture. For the ImageCaptioner+, we fix the ImageCaptioner models and only fine-tune the language model, i.e., LLaMA2-7B. We use the prediction loss of behavior tokens as the evaluation criterion, namely eval loss. We found that the model trained with LLaVA-7B architecture has the lowest evaluation loss, so we chose this model as the default model.

**Behavior Tokenizer.** We explore `OmniJARVIS` with different behavior tokenizers, including the default setting using FSQ codebook, a variant of using VQ-VAE instead of FSQ [42], and simply using sub-goal language annotation as behavior "tokens". The evaluation results on 4 programmatic tasks are listed in Table 4. Using an FSQ tokenizer is generally better than a language goal, which confirms the advantages of using a tokenized behavior over language descriptions of behavior. The use of VQ-VAE as a quantized behavior tokenizer collapsed during the training process, so there were no results in all test tasks.

**Behavior Codebook.** We conduct an in-depth investigation of behavior tokenizers with varying codebook sizes, utilizing recommended sets of FSQ levels to approximate specified codebook dimensions [34] as delineated in Table 5. We evaluate performance across multiple metrics for each codebook size. **Codebook Usage** is quantified as the proportion of codewords utilized at least once when encoding the validation datasets. **Reconstruction FSD** is measured by the FSD scores derived from the MineCLIP encoder [16], processing 1,000 different demonstration videos through the FSQ-GROOT and subsequent rollout in a randomly generated environment. Additionally, we measure **Resampling FSD**, which is the FSD score obtained when the environment rollout is conditioned on representations sampled from the codebook. Finally, we assess the **average rewards** for the task "collect wood" using `OmniJARVIS` across varying codebook sizes. Our findings indicate that increases in codebook size correlate with enhanced average rewards and reduced FSD scores, suggesting a scalable performance in `OmniJARVIS` with larger codebooks.

**Behavior Semantics.** We provide some qualitative analysis on the learned FSQ-based behavior tokenizer. In Figure 6, we tokenize several reference videos, then feed the behavior tokens to the policy decoder and see if it can accomplish the same task as in reference videos. The results indicate that our behavior tokenizer is able to capture such behavior semantics and offers rich task information.

### 4.6 Generalization and Scaling Potential of `OmniJARVIS`

We first explore adapting `OmniJARVIS` to the Atari game Montezuma's Revenge. We created a dataset from 500 episodes played by an agent trained with Random Network Distillation [8],

Provided ← | → Generated

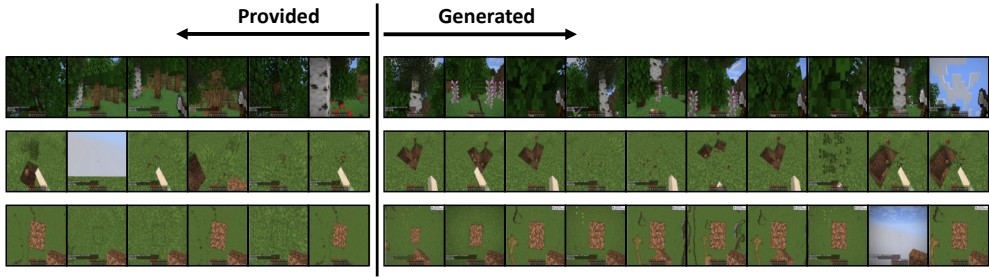

Figure 6: **Examples of behavior tokenization-detokeinzation.** Left: the reference video to be tokenized by our FSQ-based behavior tokenizer (encoder). Right: the behavior of the policy decoder is conditioned on the behavior tokens. The policy decoder can reproduce the task being accomplished in the reference video.

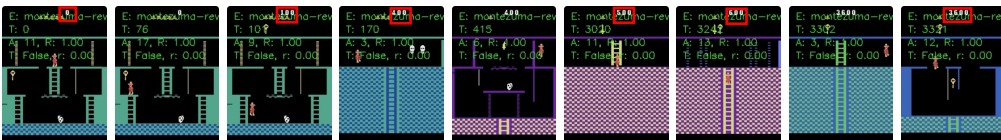

Figure 7: `OmniJARVIS` **plays Montezuma's Revenge and gets a reward of 3600.**

supplemented by random actions in early frames to enhance diversity. This dataset contains 1,823,699 transitions. We then trained the FSQ-GROOT tokenizer on this new dataset and subsequently trained `OmniJARVIS` on the tokenized data. The finetuned `OmniJARVIS` achieved a score of 3600 in Montezuma's Revenge, indicating promising transferability. A rollout trajectory is in Figure 7.

We also investigate the scaling effect [25, 30] of data and model in `OmniJARVIS` by monitoring the instruction-following loss on the validation set as the amount of data increases. In addition to fine-tuning from the default LLaVA-7B, we include two additional scales: `OmniJARVIS`-2B (fine-tuned from LLaVA-2B with Gemma-2B language models [20]) and `OmniJARVIS`-13B (fine-tuned from LLaVA-13B with LLaMA2-13B language models [31]).

The validation loss curves in Figure 5 reveal the following insights: 1) When using Omni-Tokenizer, `OmniJARVIS`'s instruction tuning aligns with the scaling law [25]. All curves exhibit a log-linear decrease as the data scale increases. 2) Scaling up VLM consistently enhances performance. Notably, `OmniJARVIS`-7B demonstrates significantly lower losses compared to `OmniJARVIS`-2B. However, while improvements are consistent, the difference between `OmniJARVIS`-7B and `OmniJARVIS`-13B seems less pronounced, hinting at potential saturation when further scaling up VLM. This underscores both the scalability of `OmniJARVIS` and the importance of increasing data volume to match the model.

## 5    Related Works

**Pretrained Language Models for Decision-making.** Several works have explored leveraging LLMs to generate action plans for high-level tasks in embodied environments [23, 27, 5, 52]. To better perform complex planning in the environment, existing methods usually utilize chain-of-thought [49] or related methods [50]. To better cope with uncertainties in open worlds, some LLM-based methods generate plans interactively with human and environmental feedback [39, 46, 24] and retrieving from memory [47] or internet corpus [48]. However, those plans can only be executed in a language environment or require an additional controller or code executor to interact in an open world.

**Vision-Language-Action Models.** In order to better utilize the knowledge inside the language model for decision-making, some methods tend to use decision datasets to fine-tune pretrained language models [15, 14]. Gato [38] was among the first to tokenize environment-provided actions to enable joint sequential modeling across modalities. PaLM-E [14] generates high-level instructions as texts and uses dedicated controllers to perform the task described by the output instructions. The RT series focuses more on robotics settings. Specifically, RT-1 pairs a VLM with a language-conditioned controller; RT-2 extends the VLM to directly include control tokens; RT-X generalizes to new robots and environments. A recent VLA model LEO [22] expands the perception from 2D images to 3D world and enables rich scene-level reasoning and control tasks.

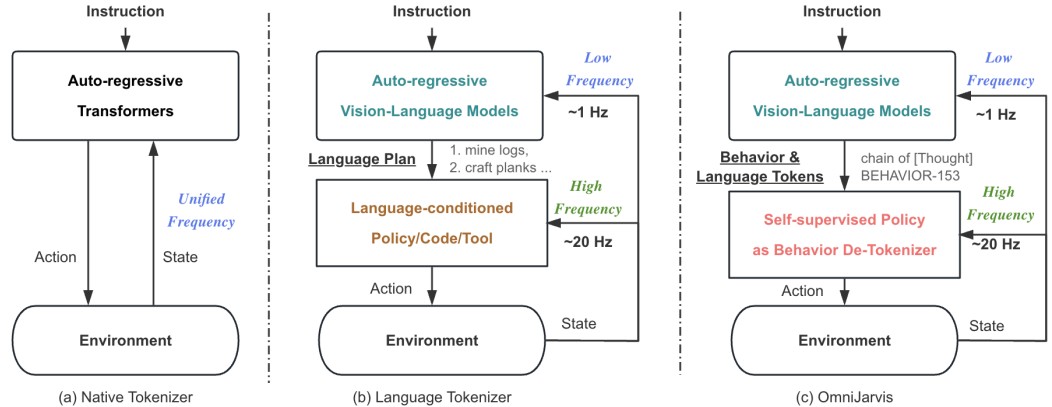

Figure 8: **Comparative Framework of Vision-Language Action Models. (a)** depicts a model where upon receiving a language instruction, actions are directly output based on the environmental state, facilitating immediate interaction with the environment at a unified frequency. Smaller models with <1B parameters like VPT [2] maintain higher frequencies (>20Hz), though their capability for complex reasoning tasks is limited. Larger models with >7B parameters such as RT-2 [6], offer enhanced performance but operate at significantly reduced frequencies (2-3Hz). **(b)** illustrates a common approach utilizing large vision-language models for planning, subsequently outputting language goals [46, 14, 4]. A language-conditioned policy then translates these language goals into actions at a real-time interaction rate of 20Hz, with high-level models re-planning at less than 1Hz. This hierarchical structure balances interaction frequency and performance, while it requires language as an intermediary and additional language labels. The training process of high-level vision-language models and language-conditioned policies are separate, thus performing poorly on tasks that can not be easily connected by language. **(c)** (ours) mirrors the hierarchical structure of (b) but differentiates by employing a self-supervised encoder-decoder policy [10] and FSQ quantization [34] as a behavior tokenizer. The upper-level vision-language models produce self-supervised behavior tokens, which are then conditioned by a policy decoder to output actions, facilitating environment interaction. The behavior tokens are injected into the training corpus of vision-language-action models, which enables end-to-end inference. This approach also eliminates the need for external language supervision and scales efficiently.

**Open-world Agents in Minecraft.** As LLMs have achieved remarkable reasoning results and understanding capabilities across various domains, the year 2023 has witnessed researchers adopting multiple LLM-based approaches to create open-world agents in Minecraft [46, 55, 47, 44]. Some methods focus on building policies for low-level skills [10, 28, 2]. Building upon the low-level policies to interact with the Minecraft environment, Wang et al. [46], Yuan et al. [51] and Wang et al. [47] focus on leveraging the pre-trained language models as planners to finish programmatic tasks with in-context learning. Wang et al. [44] adopts the life-long learning scheme and generates code as policies to enable continual exploration. Some use expert trajectories and Minecraft corpus to fine-tune pre-trained vision language models for better embodied planning [36, 54].

# 6 Conclusion

We've presented `OmniJARVIS`, a novel VLA model that encompasses strong reasoning and efficient decision-making capabilities via unified tokenization of vision, language, and actions in multimodal interaction data. The key ideas are learning behavior tokenizer (trajectory encoder) and de-tokenizer (IL policy decoder) using self-supervised learning on behavior trajectories and autoregressive modeling of tokenized multimodal interaction data using a pretrained multimodal language model (MLM). Evaluations on the open-world Minecraft Universe demonstrate its impressive instruction-following capabilities. Possible future directions include a more in-depth investigation of behavior tokenization, language capabilities after VLA fine-tuning, and alignment concerns emerging from the unified interaction modeling and VLA capabilities.

## Acknowledgments

This work is funded in part by the National Key R&D Program of China #2022ZD0160301. We thank a grant from CCF-Tencent Rhino-Bird Open Research Fund. One author is funded in part by NSF grants #IIS-1943641, #IIS-1956441, #CCF-1837129, an SRA from Meta and a research gift from Amazon Alexa AI, and a gift from RelationalAI.

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

# A  Training Details

`OmniJARVIS`. We utilized the SFTTrainer class from the TRL library by Hugging Face to train the VLM model. The learning rate was set at 1.4e-5, and a cosine learning rate scheduler was employed. The weight decay parameter was set to 0 with a warm-up ratio of 0.03. Training took place on 8 A800 GPUs with FSDP, with a batch size of 2 and gradient accumulation steps of 4 using bf16 precision. The training lasted for one epoch on our generated dataset. The raw interaction dataset comes from the sections 6xx, 7xx, and 10xx of the contractor dataset provided by OpenAI [2] and the recording interactions of JARVIS-1 Agents [47].

**Behavior Tokenizer.** Each frame in our experiments has a resolution of 128x128 pixels. We segmented each episode into multiple trunks, with each trunk consisting of 128 frames. The learning rate was set at 0.00004, with a weight decay of 0.001. The batch size was configured to 2, and training was conducted on a cluster of eight NVIDIA 3090 Ti graphics cards. The training dataset comprised sections 6xx, 7xx, 9xx, and 10xx of the contractor dataset provided by OpenAI [2]. The precision for training was set to bfloat16.

# B  FSD Computation

This section outlines the computation of FSD score between the generated videos and human gameplay recordings. First, we divide the videos into trunks of 128 frames. For each segment, we sample 16 frames, with 8 frames in between each sampled frame. These sequences of 16 frames are then fed through the video encoder of MineCLIP [16] to obtain 512-dimensional video embeddings. Finally, the score is calculated according to [19] between the embeddings of the generated videos and the reference videos.

We compute FSD scores between and within sets of videos using three distinct tasks, as illustrated in Figure B.1. A noticeable gap exists between the FSD scores calculated within the same set of videos and those calculated between different sets. Furthermore, the metric exhibits relative insensitivity to the number of videos used for computing the score, demonstrating the validity of our proposed metric.

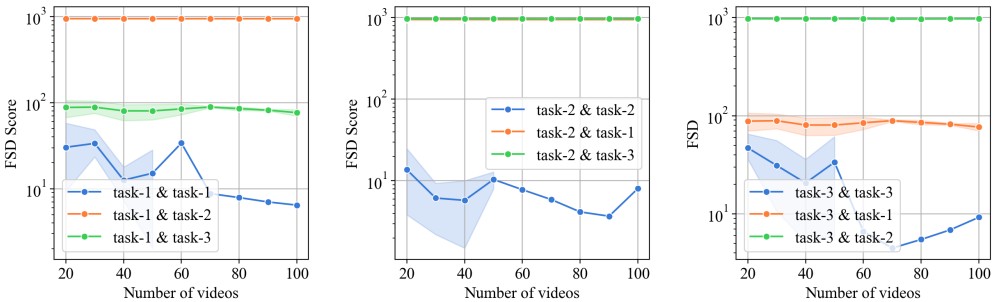

Figure B.1: FSD scores between and within sets of videos for two distinct tasks. The horizontal axis represents the number of videos used for computing the scores, and the vertical axis depicts the corresponding score.

# C  Benchmarks Details

## C.1  Programmatic Tasks

Table 6: Description and setting of programmatic tasks.

| Group | Task Number | Maximum Steps | Task Prompt | Item List |
|---|---|---|---|---|
| Wooden | 10 | 3000 | Make {item} from empty inventory. | stick, crafting_table, chest, ladder, bowl, button, door, boat, ... |
| Food | 5 | 6000 | Get the food {item}. | cooked_chicken, cooked_mutton, cooked_porkchop, cooked_beef, bread |
| Stone | 5 | 3000 | Craft a {item}. | charcoal, smoker, stone_sword, furnace, torch |
| Iron | 5 | 6000 | Smelt iron ingots and craft {item}. | iron_sword, iron_ingot, bucket, iron_nugget, shears |
| Diamond | 5 | 12000 | Dig down and craft {item}. | diamond_pickaxe, diamond_shovel, diamond_hoe, diamond_axe, diamond |

Table 7: Embodied Question Answering Examples.

| Category | Question | Answer |
|---|---|---|
| Planning | How to obtain bucket with empty inventory step-by-step in Minecraft? | 1. mine 4 log without tool. . . |
| Planning | How to obtain cooked beef with empty inventory step-by-step in Minecraft? | 1. kill cow to obtain 1 beef. . . |
| Knowledge | How many materials do I need to collect to make 2 iron ingots in one go? | To make 1 iron ingot, you need 1 iron ore and. . . |
| Knowledge | What are the materials to make 1 diamond pickaxe in Minecraft? | 3 diamond, 2 stick. |
| Knowledge | What are the materials to make 1 iron helmet in Minecraft? | 5 iron ingots. |
| Knowledge | What are the materials to make 1 golden axe in Minecraft? | 3 gold ingot, 2 stick. |
| Knowledge | What are the materials to make 1 wooden shovel in Minecraft? | 1 planks, 2 stick. |
| Knowledge | What are the materials to make 1 bread in Minecraft? | 3 wheat. |
| Reasoning | Can diamond be mined with stone pickaxe in Minecraft? | No. Diamond can only be mined with iron. . . |
| Reasoning | Can coal be mined with an iron pickaxe in Minecraft? | Yes. Coal can be mined. . . |
| Reasoning | Can obsidian be mined with an iron pickaxe in Minecraft? | No. Diamond can only be mined with iron. . . |
| Reasoning | Can lapis lazuli be mined with a diamond pickaxe in Minecraft? | Yes. Lapis lazuli can be mined |
| Reasoning | Can emeralds be mined with a stone pickaxe in Minecraft? | No. Emeralds can only be mined with an iron. . . |
| Decision-making | <image><inventory> To mine diamonds, what behavior should I do now? | craft iron pickaxe |

## C.2 Embodied Question Answering Benchmarks

The embodied question-answering benchmarks consist of questions and instructions for Minecraft benchmarks, consisting of over 100 questions on knowledge question answering, embodied planning, and math reasoning.

To enhance the LLM-based agent's understanding of Minecraft, we gathered a large dataset called Minecraft-Embodied-Question-Answering-300k (**mc-eqa-300k**). Initially, we extracted language corpus from the Minecraft Wiki[2] and DigMinecraft[3] website to create prompts for the language model (ChatGPT). This resulted in around 50k question-answer pairs. Using these as seed questions, ChatGPT generated approximately 250k diverse questions on various aspects of Minecraft. To ensure accurate responses and avoid hallucinations common with existing models like self-instruct [45], we employed a retrieval augmented generation [26] method known as RAT [48] to generate accurate answers. The resulting question-answering dataset, along with interaction data, form our training dataset for `OmniJARVIS`.

---

**Example of Embodied Question Answering Generation for Reasoning Group**

**question**: Can stone pickaxe be used to mine diamonds?
**answer**: No, a stone pickaxe cannot be used to mine diamonds in Minecraft. An iron pickaxe or better is required to mine diamonds.
**question**: Can golden pickaxe be used to mine diamonds?
**answer**: In Minecraft, you cannot use a golden pickaxe to mine diamonds. Diamonds are one of the hardest materials in the game and require an iron pickaxe or better to mine. Using a gold pickaxe on diamond ore blocks will cause them to break, but you won't receive any diamonds as a drop. You'll just get the diamond ore block itself.
To successfully mine diamonds, you should use an iron, diamond, or netherite pickaxe. These pickaxes have the required hardness to break diamond ore blocks and yield diamonds when mined.

---

[2]https://minecraft.wiki/
[3]https://www.digminecraft.com/

## Example of Embodied Question Answering Generation for Planning Group

**question**: Give you nothing in the inventory, how to obtain diamond pickaxe from empty inventory, generate a step-by-step plan, start from punching trees, and describe every step in detail.

**answer**: Obtaining a diamond pickaxe in Minecraft involves several steps, starting from punching trees and eventually progressing to mining diamonds. Here's a step-by-step plan to obtain a diamond pickaxe from an empty inventory:

**Step 1: Punching Trees**

1. **Gather Wood:** Start by punching trees to collect wood blocks. You'll need at least 3 logs to proceed.

**Step 2: Crafting Basic Tools**

2. **Craft Wooden Planks:** Open your inventory and turn the logs into wooden planks by placing them in the crafting grid (4 wooden logs -> 16 wooden planks).

3. **Craft Crafting Table:** Use the wooden planks to craft a crafting table. Place 4 wooden planks in the 2x2 crafting grid to create it.

4. **Craft Wooden Pickaxe:** With the crafting table, craft a wooden pickaxe using wooden planks and sticks. Place 3 wooden planks in the top row and 2 sticks in the middle row.

**Step 3: Finding Stone**

5. **Mine Stone:** Use your wooden pickaxe to mine stone blocks. You'll need at least 3 stone blocks to proceed.

**Step 4: Upgrading Tools**

6. **Craft Furnace:** Use the crafting table to create a furnace using 8 pieces of cobblestone. Place the cobblestone in all squares of the crafting grid except the center square.

7. **Smelt Cobblestone:** Use the furnace to smelt the cobblestone into smooth stone. Place the cobblestone in the top slot of the furnace and fuel (wooden planks, sticks, etc.) in the bottom slot. Collect the smooth stone when it's ready.

**Step 5: Crafting a Stone Pickaxe**

8. **Craft Stone Pickaxe:** With the smooth stone, craft a stone pickaxe using cobblestone and sticks. Place 3 cobblestones in the top row and 2 sticks in the middle row.

**Step 6: Upgrade to Iron Pickaxe**

9. **Find Iron Ore:** Find iron ore blocks underground (usually at lower depths) and mine them using your stone pickaxe. You'll need at least 3 iron ore blocks.

10. **Smelt Iron Ore:** Place your iron ore in the top slot of the furnace and use wooden planks as fuel in the bottom slot. This will smelt the iron ore into iron ingots.

11. **Craft Iron Pickaxe:** With the iron ingots, craft a iron pickaxe using iron ingots and sticks. Place 3 iron ingots in the top row and 2 sticks in the middle column.

**Step 7: Mining Diamonds**

12. **Find Diamonds:** Use your iron pickaxe to mine for diamonds. Diamonds are found deep underground, usually between levels 5 to 12. They appear as blue, sparkly ore blocks.

**Step 8: Crafting a Diamond Pickaxe**

13. **Craft Diamond Pickaxe:** Once you have collected at least 3 diamonds, craft a diamond pickaxe using diamonds and sticks. Place 3 diamonds in the top row and 2 sticks in the middle row.

**Step 9: Using the Diamond Pickaxe**

14. **Mine Ores:** The diamond pickaxe is more durable and efficient than lower-tier pickaxes. Use it to mine a variety of ores, including iron, gold, redstone, and more.

And that's it! You've successfully obtained a diamond pickaxe from an empty inventory by following these steps. Remember that Minecraft's world generation and mechanics might vary slightly depending on the version you're playing, so adapt the plan as needed.

## Example of Embodied Question Answering Generation for Knowledge Group

**question**: How to craft diamond pickaxe step-by-step, describe every step in details.

**answer**: To craft a diamond pickaxe in Minecraft, you will need to follow these steps:

1. **Gather Resources:**
- Start by gathering the necessary resources to craft a diamond pickaxe. You'll need three diamonds and two sticks.
- Diamonds can be found deep underground, typically between levels 1 to 15 in the Overworld. You'll need an iron pickaxe or better to mine diamonds. Use this iron pickaxe to mine the diamond ore blocks.

2. **Crafting Sticks:**
- If you haven't already, craft sticks from wooden planks. Open your crafting table and place two wooden planks vertically to create four sticks.

3. **Crafting the Diamond Pickaxe:**
- Open your crafting table, which consists of a 3x3 crafting grid.
- Place the three diamonds in the top row.
- Place two sticks in the middle row, one in the center slot and one in the slot just below it.
- Your crafting grid should now look like this:

[D][D][D]

[ ][S][ ]

[ ][S][ ]

- [D] represents a diamond.
- [S] represents a stick.
- Once you've placed the items in the correct pattern, you'll see a diamond pickaxe appear in the result box of the crafting table.

4. **Collect the Diamond Pickaxe:**
- Click or drag the diamond pickaxe from the crafting table's result box into your inventory.

Congratulations! You now have a diamond pickaxe in your inventory. Diamond pickaxes are extremely valuable in Minecraft because they can mine most blocks faster than other pickaxes and are required to mine certain valuable resources like obsidian. Be sure to use it wisely and take care of it, as it's a valuable tool in the game.

## C.3 Open-ended Instruction-Following Tasks

**Prompts for Open-ended Instruction-following Tasks**

1. Cook the beef with a furnace and recycle the furnace. If you meet night, place and use the bed for sleeping.

2. Explore caves, mine resources, and craft items in Minecraft to progress and survive.

3. Gather resources, craft tools, and cook food in Minecraft.

4. Place a torch on the wall to light the environment. Collect and picking it up when you leave.

5. Craft an oak boat and use it to travel

6. Obtain resources for building and survival by gathering materials and farming resources.

7. Consistently interact with chests to manage inventory contents.

8. Explore and gather resources in Minecraft.

9. Collect and mine azure bluets, deal damage to mobs, and defeat mobs in the game.

10. Do the following tasks sequentially: 1. Gather oak logs and oak leaves from trees. 2. Use oak logs to create oak planks and then a crafting table. 3. Create sticks from oak planks using the crafting table. 4. Craft a wooden axe and a wooden pickaxe using sticks and oak planks. 5. Collect materials like mushrooms and brown mushrooms by mining blocks with the wooden axe. 6. Mine grass, tall grass, and stone using the wooden tools for resources.

11. Harvest sugar cane to obtain multiple sugar cane pieces.

12. Plant and consume wither roses repeatedly.

13. Harvest wheat seeds, plant them, and use the harvested wheat seeds to feed animals or craft items such as bread.

14. Trade with a villager by giving emeralds and books to receive enchanted books as well as new emeralds and books.

15. Mine ice using an iron pickaxe and pick up the ice block obtained.

16. Open a chest in the game to access or manage inventory items.

# D Prompt for Instruction Generation

**Prompt D.1:Prompt for Instruction Generation**

**Instruction**:
This is a paragraph of description of the player's gameplay in Minecraft. The caption summarizes the current environmental state and agent behavior, with the timestamp indicating which frame of the video this caption is from. Please summarize what tasks the agent completed throughout the entire video. Please guess what instruction or task the player received to exhibit such behaviors. This task should be clear and in details.
**IMPORTANT**:
DIRECTLY output the task. DO NOT repeat user input. DO NOT add additional explanations or introducement in the answer unless you are asked to.
**Observation**:
Stats minecraft.custom:minecraft.interact_with_furnace happens. Gui is open. New stats minecraft.craft_item:minecraft.cooked_beef happens. Get new item: cooked_beef*9. Get new item: stone_pickaxe*1. Stats minecraft.use_item:minecraft.stone_pickaxe happens. Stats minecraft.mine_block:minecraft.furnace happens. Stats minecraft.pickup:minecraft.furnace happens. Get new item: furnace*1. Stats minecraft.use_item:minecraft.white_bed happens. Stats minecraft.mine_block:minecraft.white_bed happens. Stats minecraft.pickup:minecraft.white_bed happens. Get new item: white_bed*1. New stats minecraft.use_item:minecraft.cooked_beef happens. Consume cooked_beef*1. **Task**:
1. Interact with a furnace to smelt cooked_beef and eat the cooked_beef. 2. Place a white_bed and sleep on it to survive the night.
**Observation**:
{observation}

## Example of Instruction Generation

Example:
**Observation**:
Consume chest*1. Stats use_item:chest happens. Consume chest*1. Stats use_item:chest happens. Consume chest*1. Stats use_item:chest happens. Consume chest*1. Stats use_item:chest happens. Consume item: chest*1. Stats use_item:chest happens. Stats custom:open_chest happens. Open Game 2D GUI. Consume oak_planks*24. Consume item: birch_planks*5. Stats custom:open_chest happens. Open Game 2D GUI. Consume lapis_lazuli*22. Consume item: iron_ingot*18. Consume item: potato*30. Consume item: carrot*9. Consume item: wheat*4. Stats custom:open_chest happens. Consume oak_planks*64. Consume item: oak_planks*44. Stats custom:open_chest happens. Open Game 2D GUI. Consume item: granite*20. Stats custom:open_chest happens. Open Game 2D GUI. Consume item: oak_sapling*2. Consume item: birch_sapling*4. Consume item: wheat_seeds*12. Consume item: poisonous_potato*1. Consume item: bread*1. Stats custom:open_chest happens. Open Game 2D GUI. Stats custom:open_chest happens. Get new item: wheat*4. Get new item: carrot*9. Get new item: potato*30. Stats custom:open_chest happens. Open Game 2D GUI. Consume item: wheat*4. Consume item: carrot*9. Consume item: potato*30. Get new item: potato*15. Stats custom:open_chest happens. Open Game 2D GUI. Consume item: lapis_lazuli*64. Stats custom:interact_with_furnace happens. Open Game 2D GUI. Consume item: potato*15. Stats custom:open_chest happens. Open Game 2D GUI. Get new item: iron_ingot*18. Stats custom:interact_with_crafting_table happens. Open Game 2D GUI. Consume item: stick*28. Get new item: stick*22. Consume item: iron_ingot*18.
**Generated Instruction**:
Task: Organize and manage inventory by storing items in chests, crafting various items, and using the furnace to smelt resources efficiently. Also, focus on gathering resources like wood, ores, food items, and plant materials for crafting and survival purposes.

# E  Prompt for Thought Generation

---

**Prompt E.1:Prompt for Thought Generation**

A player is playing Minecraft. I need you to give thought about what behavior it should take next given current situation. Here are some demonstrations:

Task: "Obtain a diamond"
State: "The image captures a scene from the popular video game, There is a grass block in front of the agent."
Inventory: {'dirt': 10}
Memory: {}
Thought: "The player has nothing in its inventory, it should first go to harvest some oak logs. However there is a grass block in front of the agent, it should mine it first to find a way. So the behavior should be {mine_block:grass:1}."
Behavior: {'mine_block:grass': 1}

Task: "Obtain an iron pickaxe"
State: "In the image, a player in the video game is standing in a dark cave."
Inventory: {'dirt': 20, 'stick': 10, 'iron_ore': 5, 'furnace': 1, 'stone_pickaxe': 1}
Memory: {'mine_block:iron_ore': 5, 'craft_item:furnace': 1}
Thought: "The player has 5 iron ores in its inventory, it should smelt them to get iron ingots. However, it does not have enough coal to smelt the iron ores. The player should mine some coal ores first. And using the stone pickaxe in the inventory can help to mine the coal ores. So the behavior should be {use_item:stone_pickaxe:1, mine_block:coal_ore:1}."
Behavior: {'use_item:stone_pickaxe': 1, 'mine_block:coal_ore': 1}

Task: "Harvest logs"
State: "The image captures a moment in the video game The player's character, standing in the center of the frame, is holding a crafting table in their hands. The crafting table, which is the main focus of the image, is gray and has a crafting grid on top of it."
Inventory: {'oak_log': 20, 'stick': 8}
Memory: {'use_item:wood_axe': 40, 'craft_item:stick': 8, 'mine_block:oak_log': 20, 'break_item:wood_axe': 1}
Thought: "The player needs to harvest more logs. However the last memory shows that the player has broken the wood axe. So the player should craft a new wood axe first. The player has enough logs in the inventory and the crafting table in the inventory can help to craft the wood axe. So the behavior should be {use_item:crafting_table:1}."
Behavior: {"use_item"crafting_table": 1}

In demonstrations, "Task" is the goal of player. "State" describes the image the player is facing, "Inventory" is its current inventory and "Memory" contains past behaviors taken by the player. "Memory" is sorted by time, with the most recent behavior at the end. You should pay attention to recent behaviors. According to these information, players first generate thoughts about what to do next("Thought") and then take behaviors accordingly("Behavior"). In the first demo, the behavior is not relevant to the task; in the second demo, the thought considers needs of the task and current inventory; the third demo considers the memory of the player and identify the need of crafting a new tool to replace the broken tool. Now think about the following situation:

Task: {task}
State: {state}
Inventory: {inventory}
Memory: {memory}
Thought: {}
Behavior: {behavior}

Given current situation and the behavior the player will take, output a simple thought that will directly lead to this behavior. Please carefully revise the need of the task, current inventory and recent memory of the player. Be sure to explain every part of the behavior. The output format should be "Thought: reason...So the behavior should be {behavior}".

---

## Example of Thought Generation

Example:

Task: "The player was instructed to mine various resources and craft tools in Minecraft: 1. Start by mining coal ore and crafting cooked beef from it. 2. Smelt iron ore and cook food in the furnace. 3. Mine stone to collect cobblestone. 4. Craft a stone pickaxe and use it to mine various ores like coal, iron, and diorite. 5. Create torches from coal and sticks. 6. Craft a stone pickaxe and an iron pickaxe. 7. Use the iron pickaxe to mine granite and gather resources. 8. Interact with a crafting table to craft items like an iron pickaxe, torches, and iron ingots. 9. Utilize tools like pickaxes to mine stones and different ores efficiently. 10. Gather various resources like coal, iron, cobblestone, diorite, and granite. 11. Keep crafting and mining to progress in the game.These actions showcase a cycle of resource gathering, processing, and crafting to advance the player's capabilities and inventory in the game."

State: "The image captures a moment in the video game Minecraft. The player's character, standing in the center of the frame, is holding a crafting table in their hands. The crafting table, which is the main focus of the image, is gray and has a crafting grid on top of it. In the crafting grid, there are several items arranged in rows and columns. Starting from the top left, there's a book, followed by a loom in the middle, and a furnace at the bottom. The crafting table is set against a black background, which contrasts with the gray color of the table and the items on it. At the bottom of the image, there's a red banner with the text "Crafting" written on it. This banner adds a pop of color to the otherwise monochrome image. The overall composition of the image suggests that the player is in the process of crafting something, possibly a book or a loom, using the items in the crafting grid."

Inventory: {'wooden_shovel': 1, 'wooden_axe': 1, 'cobblestone': 51, 'dirt': 14, 'andesite': 23, 'iron_ore': 7, 'coal': 31, 'stick': 54, 'birch_log': 5, 'birch_planks': 47, 'furnace': 1, 'crafting_table': 1, 'granite': 4, 'diorite': 7, 'wooden_pickaxe': 1}

Memory: {use_item:stone_pickaxe': 63, 'mine_block:coal_ore': 9, 'pickup:coal': 9, 'mine_block:wall_torch': 1, 'use_item:torch': 3, 'pickup:torch': 1, 'mine_block:granite': 4, 'pickup:granite': 4, 'mine_block:stone': 38, 'pickup:cobblestone': 37, 'mine_block:diorite': 8, 'pickup:diorite': 7, 'use_item:cobblestone': 3, 'mine_block:iron_ore': 3, 'pickup:iron_ore': 3, 'break_item:stone_pickaxe': 1}

Updated Inventory:{'wooden_shovel': 1, 'wooden_axe': 1, 'cobblestone': 51, 'dirt': 14, 'andesite': 23, 'iron_ore': 7, 'coal': 31, 'stick': 54, 'birch_log': 5, 'birch_planks': 47, 'furnace': 1, 'granite': 4, 'diorite': 7, 'wooden_pickaxe': 1}

Updated Memory: {use_item:stone_pickaxe': 63, 'mine_block:coal_ore': 9, 'pickup:coal': 9, 'mine_block:wall_torch': 1, 'use_item:torch': 3, 'pickup:torch': 1, 'mine_block:granite': 4, 'pickup:granite': 4, 'mine_block:stone': 38, 'pickup:cobblestone': 37, 'mine_block:diorite': 8, 'pickup:diorite': 7, 'use_item:cobblestone': 3, 'mine_block:iron_ore': 3, 'pickup:iron_ore': 3, 'break_item:stone_pickaxe': 1 'use_item:crafting_table'}

Generated Thought:

Thought: The player's current inventory includes various resources like cobblestone, coal, iron ore, and granite, along with essential tools like a wooden pickaxe, shovel, and axe. Recent memory indicates the player has been actively mining various blocks and collecting resources, including coal, granite, and iron ore. However, the memory also shows that the player's stone pickaxe has broken. To progress efficiently and craft necessary items, such as torches and better pickaxes, the player should utilize the crafting table. So the behavior should be {'use_item:crafting_table': 1}.

# F   Prompt for Memory Summarization

> **Prompt F.1: Prompt for Memory Summarization**
>
> A player is playing Minecraft.The situation of the player contains 4 parts: task, state, inventory and memory. Under this situation, the player will take a behavior. And after this behavior, the player's memory will be updated to "Updated Memory". I need you to give a subpart of the player's updated memory that is most relevant to its task.
> Task is the goal of the player. State describes the image the player is facing, Inventory is its current inventory of items. Memory contains its past behaviors, each item in memory is its past behavior and the number of this behavior. The memory is sorted by time, with the most recent behavior at the end. There are mainly 9 types of behavior:
> + 'craft_item:x' means to craft an item x;
> + 'drop:x' means to drop an item x;
> + 'use_item:x' means to use an item x;
> + 'pickup:x' means to pickup an item x;
> + 'custom' means to custom its playing status;
> + 'mine_block:x' means to mine a block x;
> + 'kill_entity:x' means to kill an entity x;
> + 'entity_killed_by:x' means the player is killed by an entity x;
> + 'break_item:x' means an item x got broken.
> Here is the player's current situation:
>
> Task: {task}
> State: {state}
> Inventory: {inventory}
> Behavior: {behavior}
> Updated Memory: {updated_memory}
>
> I need you to summarize what the player has done to complete the task according to the updated memory. Please make sure every part in your summary is relevant to the task. The output format should be: "The player first ..., then ..., and finally ..." Then in a new line, try to summarize which stage of the task the player is in according to the memory.

---

**Example of Memory Summarization**

Example:

Task: "Gather various resources including andesite, granite, diorite, coal, iron ore, and cobblestone using a stone pickaxe. Craft and use torches for illumination. Upgrade from a wooden to a stone pickaxe and craft a stone sword for defense. Explore and mine in a systematic way, ensuring to light up the environment with torches and replacing tools as they wear out."

State: "The image captures a moment in a video game, specifically Minecraft. The scene is set in a dimly lit cave, with a wooden pillar standing prominently in the foreground. The player's inventory and score are displayed in the top left corner of the screen, providing a glimpse into the player's progress in the game. In the bottom right corner, the player's health and hunger bars are visible, indicating the player's current status in the game. The rest of the screen is filled with a series of lines of text, each line representing a command or instruction from the game. These commands seem to be related to the player's movement and interaction with the environment, guiding the player through their adventure in Minecraft. The image is a snapshot of a complex digital world, where every command and action is carefully calculated and executed. It's a testament to the immersive and engaging nature of video games like Minecraft."

Inventory: {'stone_pickaxe': 1, 'wooden_axe': 1, 'oak_log': 8, 'stone_sword': 1, 'andesite': 8, 'coal': 13, 'oak_planks': 2, 'charcoal': 2, 'torch': 27, 'dirt': 1, 'furnace': 1, 'cobblestone': 108, 'stick': 21, 'iron_ore': 2, 'granite': 15, 'diorite': 9, 'wooden_pickaxe': 1} Behavior: {'use_item:wooden_axe': 1, 'mine_block:oak_fence': 1}

Updated Memory: {'pickup:dirt': 1, 'mine_block:oak_leaves': 4, 'mine_block:andesite': 8, 'pickup:furnace': 1, 'use_item:stone_pickaxe': 169, 'mine_block:stone': 125, 'craft_item:charcoal': 4, 'use_item:torch': 14, 'mine_block:oak_log': 21, 'mine_block:furnace': 1, 'craft_item:stone_pickaxe': 2, 'pickup:oak_planks': 2, 'craft_item:wooden_axe': 1, 'craft_item:stick': 40, 'use_item:wooden_pickaxe': 13, 'craft_item:air': 0, 'pickup:andesite': 8, 'use_item:furnace': 1, 'mine_block:grass': 1, 'mine_block:crafting_table': 2, 'craft_item:furnace': 1, 'pickup:crafting_table': 2, 'pickup:coal': 21, 'mine_block:grass_block': 1, 'craft_item:torch': 40, 'pickup:oak_log': 23, 'pickup:cobblestone': 124, 'craft_item:oak_planks': 36, 'craft_item:crafting_table': 1, 'use_item:crafting_table': 3, 'use_item:wooden_axe': 18, 'craft_item:wooden_pickaxe': 1, 'mine_block:coal_ore': 21, 'mine_block:granite': 15, 'pickup:granite': 15, 'drop:stone_pickaxe': 1, 'pickup:stone_pickaxe': 1, 'mine_block:diorite': 9, 'pickup:diorite': 9, 'break_item:stone_pickaxe': 1, 'craft_item:stone_sword': 1, 'mine_block:iron_ore': 2, 'pickup:iron_ore': 2, 'mine_block:wall_torch': 1, 'pickup:torch': 1, 'mine_block:oak_fence': 1}

Generated Summary:

The player first gathered various resources including andesite, granite, diorite, coal, iron ore, and cobblestone using a stone pickaxe. Then, they crafted and used torches for illumination, upgraded from a wooden to a stone pickaxe, and crafted a stone sword for defense. Finally, they explored and mined systematically, ensuring to light up the environment with torches and replacing tools as they wore out.

The player is in the stage of gathering various resources, crafting torches, upgrading tools, and exploring systematically.

---

# G  Examples of `OmniJARVIS` Interaction Process

The tokens with blue color are prompted from the environment observation and user, and the tokens with red color are generated by `OmniJARVIS`.

## Example 1 of `OmniJARVIS` Interaction

**TASK:** Mine stone with an iron pickaxe, gather cobblestone, then switch to gravel, mining it with a wooden shovel to collect some. Return to mining stone with the iron pickaxe, place torches for visibility, then proceed to mine iron ore as well. The agent collects items along the way, including gravel and iron ore while constantly gathering cobblestone throughout the process.

**MEMORY:** I have mine 13 stone blocks and 1 gravel block.

**INVENTORY:** iron_pickaxe: 1, wooden_shovel: 1, wooden_axe: 1, cobblestone: 156, crafting_table: 1, andesite: 23, iron_ore: 4, coal: 31, birch_log: 5, birch_planks: 47, furnace: 1, gravel: 1, granite: 12, diorite: 28, wooden_pickaxe: 1, stone_pickaxe: 1, dirt: 14, stick: 43

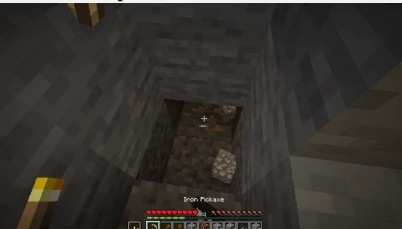

**OBSERVATION:**

**CAPTION:** The image captures a moment in the video game Minecraft. The player's character is standing at the entrance of a cave, which is characterized by its dark gray walls and ceiling. The floor of the cave is a contrasting light gray, with a few small rocks scattered about. In the bottom left corner of the screen, the player's health and hunger bars are visible, indicating the status of the character. The top right corner of the screen displays the player's inventory, showcasing a variety of items including a pickaxe, a sword, and some food. The most striking feature of the image is the text Iron Pickaxe that appears in the bottom right corner of the screen. This suggests that the player might be using an iron pickaxe, a common tool in the game for mining. The precise location of the objects and their relative positions contribute to the immersive gaming experience.

**THOUGHT:** The player needs to mine iron ore with an iron pickaxe to progress towards the task. Recent memory indicates a focus on using the iron pickaxe for various mining activities. Since iron ore is a key component in the task's objectives and the player already possesses the required tool, the next step should be to mine the iron ore. So the behavior should be 'use_item:iron_pickaxe', 'mine_block:iron_ore'.

**CONTROL:** TOKEN[BEHAVIOR-12372].

## Example 2 of `OmniJARVIS` Interaction

TASK: Task: 1. Smelt iron ingots in a furnace. 2. Craft an iron pickaxe from iron ingots and sticks. 3. Mine diamond ore blocks using the iron pickaxe, collecting diamonds. 4. Mine redstone ore, collecting redstone. 5. Mine diorite blocks. 6. Craft a diamond pickaxe using diamonds and sticks. 7. Craft additional diamond pickaxes using cobblestone and the previous diamond pickaxe.

MEMORY: I have smelt 3 iron ingots.

INVENTORY: oak_log: 9, andesite: 3, dirt: 6, oak_planks: 8, iron_ingot: 3, cobblestone: 124, stone_axe: 1, stone_pickaxe: 1, granite: 15, wheat_seeds: 1, diorite: 9, stick: 7

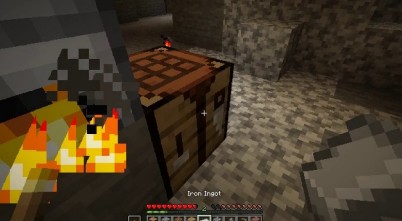

OBSERVATION:

CAPTION: The image captures a scene from the video game Minecraft. In the center of the frame, a player's hand is seen holding a torch, which is ablaze with orange and yellow flames. The torch is being used to ignite a pile of wooden planks that are stacked haphazardly on top of each other. The planks are brown and appear to be made of wood. The player's health and hunger bars are visible in the bottom left corner of the screen, indicating the player's status in the game. In the bottom right corner, the player's inventory is displayed, showcasing the tools at their disposal. The background of the image reveals a cave-like structure, characterized by stalactites hanging from the ceiling and stalagmites rising from the floor. The walls of the cave are made of gray stone, providing a stark contrast to the wooden planks. Overall, the image provides a snapshot of a moment in the game, capturing the player's interaction with the environment and the tools at their disposal.

THOUGHT: The player holds enough iron ingots in their inventory, fulfilling the task requirement to craft iron ingots in a furnace. However, to proceed with the subsequent tasks which involve mining diamond ore and redstone ore, acquiring an iron pickaxe is crucial. Considering the current inventory, the player possesses the necessary materials to craft an iron pickaxe - iron ingots and sticks. Crafting the iron pickaxe now will enable the player to move efficiently towards their goal of mining valuable resources like diamonds and redstone. Therefore, the immediate action should be to craft an iron pickaxe. So the behavior should be craft_item:iron_pickaxe.

CONTROL: TOKEN[BEHAVIOR-1375].

