# OpenReview forum: "OmniJARVIS: Unified Vision-Language-Action Tokenization Enables Open-World Instruction Following Agents"
_NeurIPS.cc/2024/Conference — NeurIPS 2024 poster_

### Official Review · Reviewer_YroM · 2024-07-11

**Soundness:** 2
**Presentation:** 3
**Contribution:** 3
**Rating:** 6
**Confidence:** 4

**Summary:**

The paper introduces a novel approach for behavior tokenization in agent domains, utilizing a unified token transformer. The key contributions include the development of a self-supervised behavior encoder that learns a vocabulary of actions. The generated discrete tokens for actions, augments vocabularies into MLMs for autoregressive modeling. The authors curate a large-scale Minecraft and multimodal QA dataset, including synthetic annotation generation, to train their model. The experiments demonstrate that the model surpasses baselines which go through a text bottleneck or directly map from pixels to low level actions.

**Strengths:**

- The introduction of a quantized action codebook and the use of FSQ for behavior tokenization represent a novel approach to representing sub-goals, as opposed to text subgoals which may be limiting or require a predefined set.
- The authors beat strong baselines DEPS and GROOT on long-horizon tasks in Minecraft, and show strong performance in open-ended instruction following and question answering.
- The paper is generally clear and well-organized.
- The proposed approach has potential applications beyond Minecraft, offering insights into behavior tokenization and hierarchical learning.

**Weaknesses:**

- The impact of the new dataset compared to the proposed architecture is unclear. Further analysis is needed to isolate the effects of the dataset from the architectural innovations.
- Given the focus throughout the paper on the architecture contribution, the paper lacks comprehensive ablations to validate the contributions of individual components, such as FSQ and curated dataset.

**Questions:**

1. The paper proposes a new dataset and architectural methods. Can you clarify the individual contributions of each and their combined impact?
2. More comprehensive ablations to demonstrate the utility of the proposed architecture and the new dataset would strengthen the paper. Additionally, comparing against multimodal LLM training methods like QFormer or Perceiver to understand the advantages of the encoder approach and training stages.
2. The ablations in Table 6 need more detail. Does the language goal include memory and caption text?
3. How many episodes per task are used in Table 6? Why not ablate the full subtasks to provide a larger sample size? Additionally, why are ablations of the training phases and behavior tokenizer architecture not included?
4. Why are Jarvis-1 and Voyager not included in the comparisons in Table 2?
5. Can the codebook be used to derive interpretable skills?

**Limitations:**

The authors address some limitations, but additional suggestions for improvement include:
- More comprehensive ablations to demonstrate the utility of the proposed architecture and the new dataset would strengthen the paper. Comparing against multimodal LLM training methods like QFormer or Perceiver could highlight the advantages of the proposed approach.
- Ensuring consistent training data across baselines would provide a clearer comparison of model performance, or providing more experiments on the contributions of each component to back up paper claims.

---

> ### Author Rebuttal · Authors · 2024-08-07
>
> > W1, Q1, and L2: Impact of interaction dataset and model architecture.
>
> Thank you for your insightful comments. To clarify the individual contributions and combined impact of the new dataset and architectural methods, we conducted specific ablation studies detailed in our paper.
>
> **Dataset Contributions**: OmniJARVIS utilizes two key datasets: a text QA dataset based on Minecraft knowledge and the Contractor dataset of human gameplay in Minecraft. The text QA dataset enhances OmniJARVIS’s understanding of Minecraft knowledge and high-level planning for complex tasks. The Contractor dataset, on the other hand, improves mastery of basic Minecraft skills such as “mine oak log.” We explored the impact of these datasets on different task types through new ablation experiments presented in **Table 2 of the supplementary PDF**. Results indicate that the QA dataset significantly influences OmniJARVIS’s performance on long-horizon programmatic tasks, while the Contractor dataset facilitates success in short-horizon atom tasks.
>
> **Architectural Innovations**: To assess the contributions of architectural innovations, we also conducted experiments on the effects of the vision encoder (Fuyu, LLaVA, isolated Captioner), behavior tokenizer (VQ-VAE, FSQ, Language), and various scales and architectures of the language model (Gemma-2B, LLaMA-7B, LLaMA-13B). These are detailed in Tables 4 and 6, and Figure 4 of the paper.
>
> Our findings demonstrate that both the datasets and architectural enhancements significantly contribute to OmniJARVIS’s ability to perform a wide range of tasks, from simple atom tasks to complex programmatic challenges. The synergy between the diverse datasets and innovative architectural elements is key to the robust performance of OmniJARVIS across different task domains.
>
> > W2: Comprehensive ablations on FSQ and dataset.
>
> Thank you for your comments. We’ve conducted several ablation studies to validate the individual contributions of our architecture:
>
> 1.	FSQ Ablations: We’ve added new experiments on FSQ codebook size and code context length, detailed in the PDF-Table-1 and **General Response**.
>
> 2.	Dataset Ablations: We add new dataset ablation in **Table 2 of the supplementary PDF** to investigate the effectiveness of the QA and interaction dataset. Detailed in Table 5 of the original paper, these experiments evaluate the impact of different dataset segments on OmniJARVIS’s performance.
>
> 3.	Behavior Tokenizer Ablations: Experiments on different tokenizer types (language, VQ-VAE, FSQ) are added to **Table 3 of supplementary PDF**, highlighting their effects on behavior generation.
>
> These studies rigorously assess the influence of each component on the model’s effectiveness.
>
> > Q2 and L1: Ablations on multimodal LLM.
>
> Thank you for the suggestion to include more comprehensive ablations and comparisons with multimodal LLMs. We have conducted experiments using various visual encoders, including FUYU (patch encoder), LLAVA (ViT), and an independent vision captioning model (ShareCaptioner+). The results of these experiments are detailed in Table 4 of the main paper.
>
> Additionally, we are currently finetuning the OmniJARVIS model using the QFormer-based BLIP2-OPT-2.7B architecture. This variant of OmniJARVIS has not yet been finished; we will update the manuscript with the results once they are available.
>
> > Q3: Details on behavior tokenizer ablation in Table 6.
>
> Thanks for your comments. This table showcases the success rates of OmniJARVIS in completing programmatic tasks using different behavior tokenizers.
>
> Regarding your specific question about language goals, they indeed include memory and caption text derived from the meta-information in the contractor data. These are converted into language goals that serve as decoder policy prompts for the language-conditioned STEVE-I model. For example, a language goal could be "chop down the tree to mine oak logs."
>
> A note on the VQ-VAE tokenizer: during its training, we encountered a posterior collapse where only one code from the VQ codebook was utilized, leading to the failure of the VQ-GROOT training. The FSQ-GROOT, in contrast, represents the final setting used for OmniJARVIS.
>
> All models in this comparison, including LLaVA-7B as the base model, used the same synthetic memory, thought, and caption data to ensure fairness in our evaluations. We continue to expand the scale of the ablation experiments (more tasks and test repetitions) and have placed the latest results in **supplementary PDF Table 3**.
>
> > Q4: More evaluation times in Table 6.
>
> In Table 6, each task was evaluated 10 times. Due to the time-intensive nature of Programmatic Tasks, which require 6000-12000 timestamps to complete, this number of evaluations was deemed a practical balance between comprehensiveness and feasibility. To expand the scale of our experiments, we selected 1-2 representative tasks from each group of Programmatic Tasks and tested each 20 times. The results of this expanded testing are presented in Table 3 of the supplementary PDF, offering a larger sample size and further validation of our findings.
>
> Due to word limit restrictions, we have added more responses to the official comments, and we hope you can see them.

---

> > ### Comment · Reviewer_YroM · 2024-08-10
> > **Thank you to the authors for their comments**
> >
> > I thank the authors for addressing my clarifications and comments. My concerns have been mostly addressed with the additional experiments. Given the impact of the data used, I think the authors should be more transparent in the writing of the paper about the impact of the dataset versus proposed architecture, and how the training data differs from that used by baselines. I have raised my score to a 6.

---

> > > ### Author Response · Authors · 2024-08-11
> > >
> > > Thank you for your response and for increasing the score. We will incorporate these dataset details and additional experiments into the main text.
> > >
> > >  If you have any further questions, please feel free to discuss them with us.

---

> ### Author Response · Authors · 2024-08-07
> **More responses due to the char limitation.**
>
> > Q5: Results comparison with Jarvis-1 and Voyager.
>
> Thank you for your inquiry. Jarvis-1 and Voyager were not included in Table 2 due to their testing frameworks and controller:
>
> 1. Testing Settings: Jarvis-1 and Voyager operate under few-shot settings, utilizing multiple inference cycles and an explicit textual memory module for life-long learning. In contrast, our experiments are conducted in a zero-shot setting, focusing on each agent’s ability to follow instructions and generalize without prior exposure.
>
> 2. Controller Differences: Voyager employs a scripted action (offered by MineFlyer) and accesses privileged environmental information (voxel and lidar data), which provides it with capabilities not shared by the baseline agents in our study, which primarily use policy-based and visual perceptive controllers.
>
> These differences make direct comparisons between the models unfair and uninformative. We will clarify this in the updated manuscript.
>
> > Q6: Using Codebook to derive interpretable skills.
>
> Thanks for the suggestions. As mentioned in the **General Response**, OmniJARVIS primarily aims at offering a more compact representation of skills in VLA compared to counterparts like RT-2 that employ language annotations, which can be expensive to obtain. We commit to exploring the interpretability of our behavior codebook as part of future work.

---

### Official Review · Reviewer_NWLR · 2024-07-12

**Soundness:** 3
**Presentation:** 2
**Contribution:** 3
**Rating:** 6
**Confidence:** 3

**Summary:**

This work presents OmniJARVIS, a instruction following agent for open-world Minecraft. The agent works by learning a behavior encoder that generates behavior tokens conditioned on textual, visual and action inputs via self-supervised learning at first stage, then a multimodal interaction sequence can be packed with the learned tokenizer and a policy decoder is trained with this tokenizer autoregressively with the objective of predicting action sequence directly. The proposed method is achieve good performance on atomic action tasks, significantly better performance on programmatic tasks comparing with baselines, and better performance on open-ended tasks where instruction is given creatively. Comprehensive ablation experiments are conducted on behavior tokenizers, input modalities, and vision tokenizer.

**Strengths:**

1. The agent has a similar structure with GROOT, but replaced VAE-based approach with Finite Scalar Quantization (FSQ).
2. The agent demonstrated significant performance gain on proposed experiments over all baselines.

**Weaknesses:**

1. Since this work get insights from GROOT, a slightly more comprehensive compare and contrast is preferred, especially, it appears the difference between these two works is not limited to how to learn trajectory representation, but also about input data format.
2. Writing could be improved:
    1. The description for 2nd stage of training seems incomplete (line 175)
    2. Typos in general
    3. The description of how to handle longer trajectories (> 128) is not clear to me.

**Questions:**

1. Is OmniJARVIS trained considering programatic task? If yes, in Table 2, do other baselines also trained with programatic task data? if not,  If not, how do you handle this difference when creating those baselines?
2. Just to confirm, in Table 2, the baselines are retrain / finetuned with the data constructed in this work?
3. GROOT and OmniJARVIS have close evaluation result on atom tasks as in Table 1, but far apart results for programatic tasks, could you provide more discussion on the reason? Clarifying first two questions would be helpful for this question.
4. Figure 5. Left and right seems similar visually, more explanation and description on what those behaviors are would be helpful for understanding.

I will consider raising score if above questions are addressed.

**Limitations:**

Authors provided a discussion of scalability of the agent: Scaling law is applicable for OmniJARVIS's instruction tuning process, the eval loss exhibit a log-linear decrease as data scale for tuning increases. Scaling up VLM improves performance but saturation is observed at 7B parameters.

---

> ### Author Rebuttal · Authors · 2024-08-07
>
> > W1: difference between GROOT and OmniJARVIS
>
> As detailed in the **General Response**, our core contribution is to propose a novel Vision-Language-Action (VLA) model architecture termed OmniJARVIS, as shown in **Figure 1 of the supplementary PDF**, aiming at resolving the issues including inference efficiency and language and annotation bottleneck.
>
> GROOT is limited to short-horizon tasks (atom tasks) and **cannot accept language instructions**, as demonstrated in Tables 1 and 2 in the original paper. OmniJARVIS, on the other hand, uses the FSQ-GROOT decoder as a de-tokenizer, enabling it to generate environment-acceptable actions from discrete FSQ codes that are produced by the VLM. This allows OmniJARVIS to handle complex, long-horizon tasks (programmatic tasks) with advanced reasoning and planning abilities.
> Please let us know if you still have further concerns about the novelty and we're more than happy to assist!
>
> > W2: Descriptions for OmniJARVIS training.
>
> Sorry for the confusion. We utilized the SFTTrainer class from the TRL library by Hugging Face to train the VLM model. The learning rate was set at 1.4e-5, and a cosine learning rate scheduler was employed. The weight decay parameter was set to 0 with a warm-up ratio of 0.03. Training took place on 8 A800 GPUs with FSDP, with a batch size of 2 and gradient accumulation steps of 4 using bf16 precision. The training lasted for one epoch on our generated dataset.
>
> > W3: typos
>
> Sorry for the typos. We've revised the manuscript accordingly.
>
> > W4: How to handle longer trajectories (>128).
>
> Thank you for your question. OmniJARVIS is specifically designed to model trajectories that exceed this length, which is a limitation in models like GROOT which can only handle fixed 128-frame segments for instruction-following.
>
> Given a long trajectory(>128), we first slice it into multiple segments of 128 frames each. These segments are then encoded into action tokens, $D^{beh}$, and integrated into an interaction dataset formulated as sequences of ${D^{instruct}, D^{mem}, D^{obs}, D^{th}, D^{beh}, D^{obs}, D^{th}, D^{beh}, ...}$. OmniJARVIS, as an autoregressive transformer model, supports a maximum token length of 8k, allowing it to model sequences that contain multiple $D^{behavior}$ tokens, effectively handling long sequences beyond 128 frames.
>
> Additionally, during *training*, we employ a sliding window technique to manage sequences that exceed 8k tokens in VLM context length. In *inference*, OmniJARVIS outputs a new behavior token every 128 frames to address long-horizon tasks such as programmatic and creative tasks.
>
> > Q1 and Q2: Programmatic tasks are included in OmniJARVIS training datasets?
>
> OmniJARVIS leverages two datasets for training: a Question-answering dataset enriched with Minecraft knowledge and the Contractor dataset of human gameplay trajectories. While the instructions in the Contractor dataset don’t exclusively pertain to programmatic tasks, they support OmniJARVIS in generalizing from learned knowledge to generate new plans and behaviors in varied instructional contexts. For the comparisons in Table 2, all models, including the baselines, were trained using the same Minecraft dataset to ensure fairness in evaluating generalization capabilities across different tasks.
>
> Additionally, the baseline models in Table 2 were not retrained or finetuned with the data constructed specifically for OmniJARVIS. However, we conducted further tests, shown in Table 3, finetuning open-source models on a Minecraft-specific Question-Answering dataset derived via ChatGPT self-instruct. This helped evaluate the impact of injecting specialized Minecraft knowledge on model performance, where the finetuned models showed improved understanding, aligning their capabilities closer to those of ChatGPT.
>
> > Q3: Discussion on GROOT performance on Atom and Programmatic tasks.
>
> Atom tasks are short-horizon tasks that can usually be completed within 600 timestamps (30s). These tasks primarily assess the agent’s mastery of basic skills in Minecraft, such as “mine oak log” and “kill sheep.” In contrast, programmatic tasks are long-horizon tasks, often requiring more than 6000 timestamps (5 min) to complete. These tasks demand complex reasoning and planning. For example, the programmatic task “obtain diamond” involves multiple intermediate steps, such as mining oak logs, crafting a wooden pickaxe, and acquiring stone and iron ore.
>
> GROOT and STEVE-I models are trained on the Minecraft Contractor data for instruction-following tasks of fewer than 128 frames. As a result, they possess certain atom skill proficiency but lack complex planning and reasoning capabilities as opposed to VLA models like OmniJARVIS. Consequently, they perform well on simple atom tasks but struggle with programmatic tasks that require sophisticated planning and reasoning.
>
> > Q4: Explanation of Figure 5.
>
> Figure 5 aims to visualize the behavior produced by our FSQ-GROOT decoder (the low-level policy adopted by OmniJARVIS) when conditioned on certain behavior codes. On the left is a screenshot of the input reference video to be encoded into behavior codes, and on the right is a screenshot of the FSQ-GROOT decoder policy's rollout in a new environment.
> The behavioral similarity between left and right screenshots verifies that the code produced by the behavior tokenizer can be consistently decoded into the same behavior as the input by decoder policy, even in novel environments.

---

> > ### Comment · Reviewer_NWLR · 2024-08-12
> >
> > I think my comments are mostly addressed by author response, I think it is important to include the author responses about training details, discussion and comparison into main content in the future. I raise my score to 6.

---

> > > ### Author Response · Authors · 2024-08-12
> > >
> > > Thank you for your response and for increasing the score. We will incorporate the content including training details, discussion, and comparisons into the main text.
> > >
> > > If you have any further questions, please feel free to discuss them with us.

---

### Official Review · Reviewer_qkWa · 2024-07-13

**Soundness:** 3
**Presentation:** 3
**Contribution:** 3
**Rating:** 6
**Confidence:** 5

**Summary:**

This work introduces OmniJARVIS, which jointly reasons over visual observations, instructions, self-generated text, and actions. OmniJARVIS models actions via behavior tokens, which are discrete embeddings that are separately learned on a behavior dataset. A policy decoder converts these behavior tokens to a sequence of low-level actions. OmniJARVIS also has a pipeline for synthesizing instructions, memory to track what happened in a long trajectory, and chain-of-thought text from offline observation action data. OmniJARVIS outperforms baselines in Minecraft on atomic, programmatic, and open-ended tasks.

**Strengths:**

1. To the best of my knowledge, the behavior tokenization is a novel way to connect VLA models to actions. This approach has the advantage that the large LLM model does not need to be run for every action generation. Instead, the lighter weight policy decoder can generate a sequence of actions.

1. The pipeline of using interleaved self-generated memory with a VLA is also novel. The paper shows the value of this additional data in Table 5 and presents a scalable way to generate it for Minecraft.

1. OmniJARVIS presents a way to scale end-to-end policies for complex long-horizon tasks like Minecraft. Prior approaches like Voyager, while able to operate on long-horizon tasks, assume primitives that can be called via language. Other methods, like STEVE-1, directly output keyboard actions but struggle with long-horizon tasks (as shown in Table 2). OmniJARVIS operates directly from pixel inputs and outputs keyboard and mouse actions yet can complete long-horizon tasks with a high success rate.

1. The paper shows extensive results in Minecraft with multiple tasks in 3 task setups of atomic, programmic, and open-ended tasks. In each setting, OmniJARVIS mostly outperforms the relevant baselines.

1. OmniJARVIS can scale to larger models and datasets, as demonstrated in Fig 4.

**Weaknesses:**

1. The paper lacks many important details. Little detail is given about the encoder and policy decoder architectures (see (2) under the questions section).

1. Important behavior token ablation experiments are missing. The context length for the behavior tokens is never analyzed and only 128 is used throughout the paper. However, this could be an important setting for the behavior tokens. Additionally, the paper does not ablate the FSQ settings to determine the necessary codebook size for the behavior tokens. The behavior tokens are also only conditioned on observation sequences without actions (L92). However, this decision is never justified.

1. Experimental result details are unclear. On L215, the paper states it uses "30 programmatic tasks to evaluate the performance of different agents". But as far as I can tell, these 30 tasks are never described.

1. The value of including the question answering dataset for instruction following is not verified. It is possible including this additional training source is the primary cause of the OmniJARVIS outperforming baselines considering it constitutes a third of the examples.

1. OmniJARVIS performs worse than the GROOT baseline in collecting harder resources like wood and cobblestone (Table 1). However, I do not see this as a large weakness because OmniJARVIS is capable of programmatic tasks unlike GROOT.

1. It's unclear how the OmniJARVIS can be used beyond Minecraft. OmniJARVIS exploits that the OpenAI Minecraft data includes oracle meta information for synthesizing the instruction, memory, and thought.

1. The paper does not clearly discuss limitations.

1. Agent behavior and failure modes are not analyzed. See point (6) of my questions.

Minor:

1. Table 2 should clarify what the numbers in parentheses next to the task type mean. I assume they are the number of programmatic tasks per category.

1. A checkmark representing the training setting is _removed_ is confusing. I suggest changing to an "x" mark instead.

**Questions:**

1. In regards to weakness (6), how can the OmniJARVIS be applied to non-Minecraft domains?

1. What are the policy architectures for the encoder and policy decoder? How many parameters are there? How long are they trained for?

1. For the behavior tokens, why introduce new tokens into the LLM vocabulary as opposed to reusing infrequently used tokens in the vocabulary as in RT-2?

1. How does the paper arrive at 1T tokens on L187, given the previous token counts add up to 1B tokens?

1. In Table 5, how can OmniJARVIS work without the instruction?

1. What are the failure modes of OmniJARVIS? I am specifically interested in where OmniJARVIS fails in the programmatic tasks in Table 2. For the challenging task of Diamond, qualitatively, what behaviors does OmniJARVIS exhibit to succeed at this task?

**Limitations:**

No, the paper does not clearly state its limitations.

---

> ### Author Rebuttal · Authors · 2024-08-07
>
> > W1 and Q2: Details on encoder and decoder of behavior tokenizer.
>
> Sorry for the confusion.
> As shown in Fig. 2, the Behavior Tokenizer consists of three parts: Encoder, Decoder, and FSQ quantizer. We perform quantization based on the original GROOT, and the Encoder and Decoder use a design consistent with GROOT.
> Specifically, the encoder includes a Convolutional Neural Network (CNN) to extract spatial information from image states $s_{1:T}$ and a non-causal transformer to capture temporal information from videos. The CNN is EfficientNet-B0 networks and the non-causal transformer is built on the code from minGPT-2 removing the causal mask.
> The decoder consists of 4 identical blocks, where each block contains a Transformer-XL block and a Flamingo gated-attention dense layer to condition the decoder on FSQ code embeddings. More details can be found in GROOT paper Section 4.1 and Section 4.2.
> The fsq quantizer consists of 5 levels of [8,8,8,6,5] with a codebook size of 15360.
> The training hyperparameters of FSQ-GROOT are as follows:
> Parallel Strategy: ddp, Accumulate Gradient Batches: 8, Batch size: 2, Precision: bf16, Image size 224*224, Encoder blocks: 8, Decoder blocks: 8, Hidden dimension: 1024, Chunk size: 128, Attention memory size: 256, Optimizer: AdamW, Weight decay: 0.001, Learning rate: 1.81e-05, Warmup steps: 2k.
>
> > W2: Ablation experiments on behavior tokenizer (codebook size and context length).
>
> Thanks for your comments.
> We conduct an in-depth investigation of behavior tokenizers with varying **codebook sizes**, utilizing recommended sets of FSQ levels to approximate specified codebook dimensions in the supplementary PDF Table-1. *Codebook Usage* is quantified as the proportion of codewords utilized at least once when encoding the validation datasets. *Reconstruction FSD* is measured by the FSD scores derived from the MineCLIP encoder, processing 1,000 different demonstration videos through the FSQ-GROOT and subsequent rollout in a randomly generated environment. Additionally, we measure *Resampling FSD*, which is the FSD score obtained when the environment rollout is conditioned on representations sampled from the codebook. Finally, we assess the *average rewards* for the task “collect wood” using OmniJARVIS across varying codebook sizes. Our findings indicate that increases in codebook size correlate with enhanced average rewards and reduced FSD scores, suggesting a scalable performance in OmniJARVIS with larger codebooks.
>
> We also conduct an ablation of experiment on different context length of behavior token in PDF Table 1. We select the codebook size e14 as the default setting and explore different context lengths of 128, and 64.
> The performances under context lengths of 64 and 128 are similar. Our insight is, that although shorter context lengths may offer better behavior granularity, they also require more frequent behavior code switching during inference, resulting in a slower speed of OmniJARVIS.
>
> > W2: Behavior tokens are only conditioned on observation sequence.
>
> The reason we use observation sequence as tokenizer input is as follows: we inherit the self-supervised learning scheme from GROOT to train the behavior tokenizer, and the learning objective is to estimate missing actions from the observation-only sequence. Including actions in the input will break this learning scheme.
>
> > W3 and Minor1: Evaluation details of programmatic tasks.
>
> Sorry for the confusion. The number behind the task category is the number of tasks in this group. The evaluated programmatic tasks are taken from DEPS, and the detailed task description and task settings are shown in the supplementary PDF Table 5. All tasks are evaluated over 20 times from an empty inventory under the open-ended generated environment in Minecraft. We will add this section to the updated manuscript.
>
> > W4: Ablation experiments on the training dataset.
>
> Thank you for your comments.
> To validate the effectiveness of including the QA dataset, we conducted ablation studies (detailed in **Supplementary PDF Table 2**). These studies revealed that while OmniJARVIS performs similarly on Atom Tasks with or without the QA dataset, there is a significant performance drop in Programmatic Tasks when the QA dataset is omitted. This drop underscores the necessity of the QA dataset for enhancing reasoning and planning capabilities essential for Programmatic Tasks.
>
> We will further clarify this aspect in the updated manuscript, ensuring a comprehensive understanding of the QA dataset’s contribution to OmniJARVIS’s enhanced performance compared to baselines.
>
> > W5: Performance drop of OmniJARVIS in Table 1.
>
> Thank you for your comments. We apologize for the confusion caused by an error in Table 1; the success rate for OmniJARVIS on the stone task should be 15.8, not 5.8 as mistakenly listed. This typo has skewed the comparison.
>
> Furthermore, it’s important to note that GROOT relies on manually selected reference videos as instructions, which can significantly influence its performance. In contrast, OmniJARVIS autonomously generates behavior tokens for the policy decoder to execute, providing a more consistent and scalable approach.
> We further tested more atom tasks to compare GROOT and OmniJARVIS in the following table. Our tests show that OmniJARVIS performs comparably to GROOT across most Atom tasks, demonstrating its robustness not just in programmatic tasks but also in short-horizon atom tasks.
>
> | | log 	| dirt 	| stone 	| seeds 	| wheats 	| wool 	| llava 	|
> |-----|-----|-----|-----|-----|-----|-----|-----|
> | GROOT 	| **14.3±4.7** 	| 19.7±8.7 	| **19.0±11.3** 	| 7.3±0.6 	| 8.7±2.2 	| 1.9±1.2 	| **1.0±0.5** 	|
> | OmniJARVIS 	| 10.8±5.2 	| **20.3±9.2** 	| 15.8±2.9 	| **8.2±3.6** 	| **10.3±1.2** 	| **2.1±1.8** 	| 1.0±0.7 	|
>
> Due to word limit restrictions, we have added more responses to the official comments, and we hope you can see them.

---

> ### Author Response · Authors · 2024-08-07
> **More responses due to the char limitation.**
>
> > W6 and Q1: OmniJARVIS on other environments.
>
> Thank you for your suggestions. We have indeed begun extending OmniJARVIS to other environments, starting with Atari Montezuma’s Revenge game, where it achieved a score of 3600. This initial success illustrates the model’s potential for generalization. Details on adapting OmniJARVIS to different environments are further elaborated in the **General Response**. We are committed to ongoing efforts to expand its applicability across various domains, demonstrating its versatility and broader utility.
>
> > W7: Add limitations.
>
> Thanks for your comments. We discuss the limitations of OmniJARVIS in the **General Response**, which will be inserted in the updated manuscript.
>
> > W8 and Q6: More analysis on agent failure modes.
>
> Thank you for your inquiry regarding the failure modes of OmniJARVIS and its behaviors, particularly in challenging tasks such as obtaining a diamond. The primary failure modes can be categorized into the following:
>
> 1. **Planning Errors**: During programmatic tasks, OmniJARVIS occasionally produces incorrect thoughts or plans. For instance, it might attempt to mine stone without a wooden pickaxe. These errors stem from inaccuracies within the QA dataset and hallucinations inherent in the language model.
>
> 2. **Action Execution Failures**: The FSQ GROOT decoder (our low-level control policy) sometimes fails to translate the behavior code into actions properly, particularly with less frequently occurring fsq codes due to data imbalance in the interaction dataset.
>
> 3. **Hallucinations in perception**: Errors in visual recognition by the Llava model can lead to incorrect scene interpretations, such as mistaking a stone for iron ore. These hallucinations hinder the agent’s reasoning and decision-making processes.
>
> In the specific task of obtaining a diamond, these failure modes manifest when OmniJARVIS incorrectly plans or executes sequences due to the above issues. However, OmniJARVIS has a high success rate on finishing such tasks and correctly sequences tasks from mining basic materials to crafting necessary tools and finally obtaining the diamond, demonstrating effective integration of vision, language, and action within one unified model.
>
> > Q3: why not re-using language tokens?
>
> Thank you for your question about integrating behavior tokens into the LLM vocabulary. We employ different strategies based on the tokenizer used:
>
> 1. Reserved Tokens: When possible, we use reserved tokens from the language tokenizer for behavior tokens to maintain semantic integrity. This happens with the tokenizer of Fuyu VLM.
>
> 2. Reusing Infrequently Used Tokens: In the absence of suitable reserved tokens, we repurpose infrequently used tokens, as seen with the Llama tokenizer. This happens with the LLaVA tokenizer. We apologize for not making this clear and have updated the manuscript.
>
> Furthermore, our FSQ design compresses extensive codebook sizes into up to 35 tokens (when the FSQ setting is 8+8+8+6+5), optimizing vocabulary use without semantic loss. These approaches balance semantic preservation with efficient vocabulary management.
>
> > Q4: Confusion on training dataset tokens.
>
> Thank you for your attention to detail. The 1T tokens were indeed a typo, the correct calculation should be 1B behavior and language tokens in total. We apologize for the confusion and will correct this in the revised manuscript.
>
> > Q5: How OmniJARVIS works without instruction?
>
> Thank you for your question. In the absence of instructions, OmniJARVIS effectively models a dataset with gameplay video only -- the videos are segmented into trunks with a default length of 128, and then all trunks are converted to codes by the behavior tokenizer. OmniJARVIS is tasked to predict these codes based on the initial visual observations of the corresponding segments. The goal of this ablation is to verify the necessity of including plans, thoughts, and other means of instruction (the "language" modality) in the VLA modeling of OmniJARVIS.
>
> > Minor 2: checkmark in training settings
>
> Apologies for the confusion. We use a checkmark to indicate when the model's training data includes specific information. In Q5, the first row without an instruction represents unconditional OmniJARVIS, which is unable to follow human language instructions. With richer synthetic data, including thought, memory, and caption data, OmniJARVIS can better follow instructions, leading to improved learning with lower loss. This demonstrates the effectiveness of synthetic data. We will follow your suggestion to replace the symbols.

---

> > ### Comment · Reviewer_qkWa · 2024-08-08
> >
> > Thank you for the response and clarifications. It is essential to include all these details in the paper after the rebuttal. I also recommend directly including all details in the paper rather than referring to the GROOT for the full details. With these added details, I raised my score.
> >
> > It is encouraging to see OmniJARVIS working on Montezuma's revenge, but it's unclear how the data formation described in Section 3.1 is used in this environment.

---

> > > ### Author Response · Authors · 2024-08-09
> > >
> > > Thank you for your response and for increasing the score. We will incorporate these details into the main text.
> > >
> > > OmniJARVIS in Minecraft utilizes a dataset comprising QA datasets, synthetic instructions, memory, captions, thoughts, observations, and behavior data.
> > >
> > > However, on Montezuma's Revenge, due to time constraints, we have not yet constructed synthetic memory and thought data. Initially, we retrained the Behavior Tokenizer on Montezuma's data—specifically FSQ-GROOT—and encoded the dataset to obtain Behavior data $D^{bhv}$. The Instruction $D^{inst}$ is set as "*Play Atari Montezuma's Revenge and get a higher score.*" Observations $D^{obs}$ consist of raw image data, while Captions utilize pretrained Visual Language Models' foundational capabilities. Using this data to train OmniJARVIS; during this training process, no Memory or Thought data was incorporated yet.
> > >
> > > Despite not using complete synthetic datasets OmniJARVIS performed well in playing Montezuma’s game earning rewards up to 3600 showing good generalization across different long-horizon games . The relevant synthesis work continues, and we believe a full set of training would improve performance even more.
> > >
> > >  If you have any further questions, please feel free to discuss them with us.

---

### Official Review · Reviewer_LGfx · 2024-07-13

**Soundness:** 2
**Presentation:** 3
**Contribution:** 3
**Rating:** 7
**Confidence:** 4

**Summary:**

This paper presents a Vision-Language-Action (VLA) model, OmniJARVIS, for open-world instruction-following agents in Minecraft. OmniJARVIS leverages unified tokenization of multimodal interaction data to enable strong reasoning and efficient decision-making capabilities. This work introduces a behavior tokenizer to encode behavior trajectories into compact representations that could be effectively modeled with other modality tokens via autoregressive transformers. The experiments are conducted in a variety of tasks in the Minecraft environment, along with some analysis of design choices and scaling properties.

**Strengths:**

1. This paper is well-written and easy to follow.

2. Illustrations of the pipeline are informative and clear. The authors demonstrate their proficiency in pipeline illustrations in Fig.1 and Fig.3, which contribute to improving readability.

3. The proposed approach is scalable with respect to model sizes. As depicted in Fig. 4, a larger model results in lower evaluation loss, showcasing the scalability of this approach.

4. The authors greatly utilized established LLMs to construct a large-scale multi-modal interaction dataset. The proposed strategy of data collection could provide inspiration for future research in multi-modal embodied understanding.

**Weaknesses:**

1. Incremental design with limited novelty. To my knowledge, the key idea of this work is tokenizing observations into compact behavior representations. However, the detail of the tokenizer works similarly to the previous work GROOT. The most notable difference between the two methods is that GROOT utilized continuous latent while this work uses quantized one (from lines 88-89). In my view, the idea of quantizing the action space seems incremental to the original GROOT and cannot adequately support the innovative quality of this paper. The authors are encouraged to add a separate section/subsection to detailedly discuss the difference between these two works.

2. Lack of training/inference efficiency analysis and comparison. The authors claimed the efficiency of this approach multiple times throughout the paper, such as in lines 102-104. However, there is no direct comparison of the actual training/inference speed/memory cost between different approaches. Some quantitative supporting evidence could be helpful.

**Questions:**

Since the authors leverage GPT3.5 to augment previous datasets with more language labels, what is the price/cost for constructing such a large dataset with GPT APIs?

**Limitations:**

There’s no separate "Limitations" section in this paper. The authors are encouraged to point out that the proposed approach is only validated in the Minecraft environment, and the generalization and transferability to other scenarios are not explored in this paper.

---

> ### Author Rebuttal · Authors · 2024-08-07
>
> > W1: Novelty concerns.
>
> Thank you for your comments. As detailed in the General response, our core contribution is to propose a novel Vision-Language-Action (VLA) model architecture termed OmniJARVIS, as shown in **Figure 1 of the supplementary PDF**, aiming at resolving the issues including inference efficiency and language and annotation bottleneck. Please let us know if you still have further concerns about the novelty and we're more than happy to assist!
>
> > W2: Training/Inference efficiency analysis and comparison.
>
> Thank you for your valuable feedback. We’ve added comparisons of inference speed, memory usage, and model parameters to **Table 4 in the supplementary PDF**:
>
> **Inference Efficiency Comparison**:
> The inference efficiency experiments are conducted on the same workstation with one RTX3090 GPU.
> 1. Inference Speed: Based on Minecraft’s rendering speed of approximately 25 fps, we found that the inference speed of models using a native action tokenizer (direct action output), such as GROOT and VPT, inversely correlates with model size. The same architecture’s VPT model shows decreasing inference speeds with increasing parameters: 1x, 2x, and 3x.
> 2. Memory Usage: Models based on language tokenizers, which require in-context learning facilitated by ChatGPT, exhibited the lowest inference speeds due to their complex computational requirements.
> 3. OmniJARVIS Performance: Thanks to its self-supervised behavior tokenizer and hierarchical architecture, OmniJARVIS achieves faster inference speeds (16.62 fps) even with a larger parameter count (7.2 billion) compared to native tokenizer models like VPT-3x (0.5 billion parameters at 12.34 fps).
>
> **Training Efficiency**:
> OmniJARVIS enhances training efficiency by encoding N=128 frames of trajectory data into k=5 behavior tokens. This compression allows the model to handle data more effectively compared to models like RT-2 and OpenVLA, which predict actions directly. This results in a significant reduction in training data volume and corresponding training time.
>
> These quantitative results have been included in the revised paper to substantiate our claims regarding the efficiency of OmniJARVIS compared to other approaches.
>
> > Q: cost for synergized dataset.
>
> Due to the relatively certain generation of Thought, Memory, and Instruction, we used `gpt-3.5-turbo` to synthesize this data, and processing the Contractor dataset cost around $400.
>
> > L: Add a separate "Limitations" section and Generalization of OmniJARVIS.
>
> Thank you for your suggestion. We have added a dedicated "Limitations" section in the **General Response** to address the scope of validation for OmniJARVIS. Initially, the model was validated solely within the Minecraft environment. Recognizing the need for broader applicability, we have also conducted preliminary generalization experiments in Atari Montezuma’s Revenge environment. These results, discussed in the **General Response**, demonstrate OmniJARVIS’s potential for adaptation to different scenarios. We will provide further details on these aspects in the updated manuscript to ensure a comprehensive understanding of the model’s limitations and capabilities.

---

> > ### Comment · Reviewer_LGfx · 2024-08-12
> >
> > Thanks for the sufficient response in rebuttal, which greatly addressed my concerns. Moreover, I think the framework comparison in Fig.1 of the rebuttal pdf is clear and valuable, which should be highlighted and added to the revised paper together with other required results. Overall, I'm happy to see the paper being accepted, thus I would raise my score to 7 (accept).

---

> > > ### Author Response · Authors · 2024-08-12
> > >
> > > Thank you for your response and for increasing the score. Following your suggestions, we will add the framework comparison in Fig.1 and other experimental results to the revised paper.
> > >
> > > If you have any further questions, please feel free to discuss them with us.

---

### Author Rebuttal · Authors · 2024-08-07

Thank all. We would like to first explain some sharing concerns.

* Comparative Framework of existing Vision-Language-Action (VLA) Models.

According to the structure of the model and the training methods, the current mainstream VLA models can be roughly divided into three categories (**Figure 1 of the supplementary PDF**):
**(a)** depicts a model where upon receiving a language instruction, motor commands(actions) are directly produced based on the environmental state, facilitating immediate interaction with the environment at a unified frequency. Smaller models with <1B parameters like *VPT* maintain higher interaction speed (>20Hz), though their capability for complex reasoning tasks is limited. Larger models with >7B parameters such as *RT-2*, offer enhanced performance but operate at significantly reduced speed (2-3Hz).
**(b)** illustrates a common approach utilizing large vision-language models (VLMs) for planning, subsequently outputting language goals, e.g., *DEPS* and *PaLM-E*. A language-conditioned policy (controller) then translates these language goals into actions at a real-time interaction rate of 20Hz, with high-level models re-planning at less than 1Hz. This hierarchical structure balances interaction frequency and performance, while it requires language as an intermediary and additional language annotations. Also, the training process of high-level VLMs and language-conditioned policies are separate and thus could struggle on tasks that can not be easily depicted by language (ex. building tasks, the language descriptions can be tedious).
**(c )** OmniJARVIS(ours) mirrors the hierarchical structure of (b) but differentiates by employing a self-supervised encoder-decoder policy (GROOT) and FSQ quantization as a behavior tokenizer to connect the planner and the controller. The upper-level VLMs produce behavior tokens, which are effectively **compact** representations of tasks, and drive a policy decoder to output actions. The behavior tokens are automatically produced by tokenizing interaction data using the behavior tokenizer (see Fig.1 in the main paper), eliminating the need for language annotations on tasks/behaviors as used by the previous VLA scheme, therefore scaling more easily.

* Differences between GROOT and OmniJARVIS

Our core contribution is the construction of a brand new Vision-Language-Action structure, as shown in **Figure 1 of the supplementary PDF**. GROOT can be classified as a (a)-class model. In addition to the aforementioned characteristics, GROOT is unable to accept language instructions and can only complete short-horizon tasks (atom tasks), as shown in Tables 1 and 2 in the main paper. In OmniJARVIS, the decoder of FSQ-GROOT is used as the de-tokenizer for OmniJARVIS, which conditions on the discrete fsq code generated by OmniJARVIS to output environment-acceptable actions $a_t$ based on observation $o_t$.


* Limitations of OmniJARVIS

1. More Evaluation Environments: While OmniJARVIS’s training method does not have specific requirements for the environment or data, allowing it to potentially generalize to other environments given the appropriate data, it has so far only been tested in the Minecraft environment. Collecting data from other environments to further validate its generalizability is an ongoing work. We have finetuned OmniJARVIS in the Atari Montezuma environment and show the primary result in **Supplementary PDF Figure 2**.
2. Large-Scale Data Requirements: Training large-scale Vision-Language-Action (VLA) models like OmniJARVIS requires a significant amount of interaction data. This can be resource-intensive and may limit the feasibility of deploying such models in environments where extensive data collection is difficult.
3. Model Hallucinations: Large models like OmniJARVIS may experience hallucinations, where the model generates incorrect or nonsensical outputs. This can lead to failures in decision-making processes for certain tasks, impacting the overall reliability of the agent.


* Generalization to Other Environments

Thank you for your suggestion. The primary validation of OmniJARVIS has been within the Minecraft environment as detailed in our paper. However, extending our approach to other environments to test generalization capabilities is an ongoing work. For instance, we have recently adapted OmniJARVIS to the Atari game Montezuma’s Revenge. Here, we created a dataset from 500 episodes played by an agent trained with Random Network Distillation, supplemented by random actions in early frames to enhance diversity. This dataset contains 1,823,699 transitions.

We then trained the FSQ-GROOT tokenizer on this new dataset and subsequently trained OmniJARVIS on the tokenized data. In initial tests, the finetuned OmniJARVIS achieved a score of 3600 in Montezuma’s Revenge, indicating promising transferability. We visualize a rollout trajectory in the **supplementary PDF Figure 2**.

The training components of OmniJARVIS include a self-supervised Behavior Tokenizer for short-horizon tasks and a Vision Language Transformer that leverages a long-horizon multimodal interaction dataset. This setup aligns with formats used in other embodied datasets, suggesting potential for broader applicability.

Due to time constraints, comprehensive experiments in additional environments are planned but not yet complete. Future work will focus on demonstrating OmniJARVIS’s generalization and robustness across diverse settings.

---

### Decision · Program_Chairs · 2024-09-25

**Decision:**

Accept (poster)

**Comment:**

The paper presents OmniJARVIS, a vision-language-action (VLA) model for open-world instruction following. Integral to the model is a self-supervised behavior tokenizer that is conditioned upon text, vision, and action inputs. Behavior tokens are then decoded by a policy decoder and converted to a sequence of low-level actions. The encoder has the advantage that long-horizon sequences of instructions, observations, self-generated text, and behaviors can be combined into unified token sequences and processed via an autoregressive transformer. OmniJARVIS additionally includes a pipeline for synthesizing instructions, memory to track longer histories, and chain-of-thought text from offline observation-action data. Experiments demonstrate that OmniJARVIS outperforms contemporary baselines on a variety of tasks in Minecraft.

The paper was reviewed by four referees who agree on many of the strengths and weaknesses of the initial submission. Notably, at least two reviewers appreciate the novelty of the proposed behavior transformer as an effective means of connecting VLAs to low-level actions, with the advantage that it avoids needing to perform inference using LLMs for every action generation. Reviewers emphasized the fact that the method scales with regard to model sizes, which is not necessarily the case with existing methods. As several reviewers note, the experimental results demonstrate strong gains in performance over state-of-the-art baselines in Minecraft. Other reviewers found the paper to be well written and easy to follow, and the illustrations to be clear.

The reviewers similarly agreed regarding the weaknesses of the paper as initially submitted. These include the lack of important algorithmic and experimental details (e.g., with regard to the encoder and policy decoder architectures). Reviewers found the experimental evaluation to be lacking, particularly with regard to the need for token and context length ablations, and the need to experimentally support claims about training/inference efficiency. Additionally, reviewers questioned the differences between OmniJARVIS and the existing GROOT method as well as the nature of dataset on which the model was trained. The authors made a concerted effort to address these and other issues during the rebuttal feedback, which helped to clarify the core contributions of the work as indicated by the fact that all four reviewers increased their overall scores. As they claim to do in their response, the authors are strongly encouraged to incorporate the results of this discussion, including the additional experiments and the nature and impact of the dataset, into the paper (and minimizing the need to refer readers to the GROOT paper for details as pointed out by Reviewer qkWa).